# The Hardness Analysis of Thompson Sampling for Combinatorial Semi-bandits with Greedy Oracle

**Fang Kong**[1]     **Yueran Yang**[1]     **Wei Chen**[2]     **Shuai Li**[1*]

[1]Shanghai Jiao Tong University     [2]Microsoft Research

{fangkong,yangyr99,shuaili8}@sjtu.edu.cn     weic@microsoft.com

## Abstract

Thompson sampling (TS) has attracted a lot of interest in the bandit area. It was introduced in the 1930s but has not been theoretically proven until recent years. All of its analysis in the combinatorial multi-armed bandit (CMAB) setting requires an exact oracle to provide optimal solutions with any input. However, such an oracle is usually not feasible since many combinatorial optimization problems are NP-hard and only approximation oracles are available. An example [30] has shown the failure of TS to learn with an approximation oracle. However, this oracle is uncommon and is designed only for a specific problem instance. It is still an open question whether the convergence analysis of TS can be extended beyond the exact oracle in CMAB. In this paper, we study this question under the greedy oracle, which is a common (approximation) oracle with theoretical guarantees to solve many (offline) combinatorial optimization problems. We provide a problem-dependent regret lower bound of order $\Omega(\log T/\Delta^2)$ to quantify the hardness of TS to solve CMAB problems with greedy oracle, where $T$ is the time horizon and $\Delta$ is some reward gap. We also provide an almost matching regret upper bound. These are the first theoretical results for TS to solve CMAB with a common approximation oracle and break the misconception that TS cannot work with approximation oracles.

## 1  Introduction

Stochastic multi-armed bandit (MAB) problem [20, 5, 3] is a classical online learning framework. It has been extensively studied in the literature and has a wide range of applications [13, 12, 20]. The problem is modeled by a $T$-round game between the learning agent and the environment. The environment contains an arm set and each arm is associated with a reward distribution. At each round $t$, the agent first selects an arm, while the environment generates a random reward for each arm from its reward distribution. The agent then obtains the reward of the selected arm. The objective of the agent is to accumulate as many expected rewards over $T$ rounds, or equivalent to minimizing the cumulative expected regret, which is defined as the cumulative difference between the expected reward of the optimal arm and the selected arms over $T$ rounds. To achieve this long-horizon goal, the learning agent has to face the dilemma of *exploration* and *exploitation* in each round. The former aims to try arms that have not been observed enough times to get a potentially higher reward, the latter focuses on the arm with the best observed performance so far to maintain a high profit. How to balance the tradeoff between exploration and exploitation is the main focus of the MAB algorithms.

One of the most popular bandit algorithms is the upper confidence bound (UCB) algorithm [5]. The algorithm aims to construct confidence sets for unknown expected rewards and selects arms according to their highest upper confidence bounds. The UCB-type algorithms have been widely studied and provided theoretical guarantees with regret upper bound of order $O(\log T/\Delta)$, where $\Delta$ is the

---

*Corresponding author

minimum gap between the expected reward of the optimal arm and any suboptimal arms. Thompson sampling (TS) [3, 26] is another popular method to solve MAB problems. It is a randomized algorithm based on the Bayesian idea, which maintains an iteratively updated posterior distribution for each arm and chooses arms according to their probabilities of being the best one. The TS algorithm was introduced in the 1930s [28], but its theoretical analysis is open until recent years [15, 2]. It was proven that the regret upper bound of the TS algorithm is of the same order of $O(\log T/\Delta)$ in MAB problems [3]. Benefited from the advantages of easier implementation and better empirical performance compared to UCB, the TS-type algorithms attract more attentions in recent years.

Despite its importance, the MAB framework may fail to model many real applications since the agent's action is usually not a single arm but a combination of several arms. This motivates the study on the combinatorial MAB (CMAB) problem [11, 9, 17, 31, 7, 29, 30, 24]. In the CMAB framework, the agent selects a combination of base arms as an action to play in each round. All outcomes of these selected arms are then revealed to the agent, which is called *semi-bandit feedback* and is widely studied in the literature [7, 29, 32, 30, 14, 24]. Such CMAB framework can cover many real application scenarios including probabilistic maximum coverage (PMC) [6], online influence maximization (OIM) [29], multiple-play MAB (MP-MAB) [17] and minimum spanning tree (MST).

TS-type algorithms have recently attracted a lot of interest in CMAB problems [17, 30, 14, 24]. All of these works need an exact oracle to provide the optimal solution with sampled parameters as input in each round. However, such oracles are usually not feasible since many offline combinatorial optimization problems, such as the offline problem of PMC and OIM [16], are NP-hard and only approximation oracles are available. With an example [30] illustrating the non-convergent regret of TS with an artificial approximation oracle designed for a specific problem instance, whether TS can work well in CMAB problems with common approximation oracles is still an open problem.

One of the most common oracles for offline combinatorial optimization problems with a theoretical guarantee is the greedy algorithm. It sequentially finds the current optimal arm, or the optimal collection of multiple arms according to the structural correlations, to maximize the current total expected reward. When the termination condition is met, it will return the set of all arms found in previous steps as the solution. The termination condition is usually formulated by a constraint on the number of steps. For example, in the PMC, OIM, and MP-MAB problems, such a process is limited to continue $K$ steps. The greedy algorithm can provide approximate solutions for offline problems of PMC [8] and OIM [16], and exact optimal solutions for offline problems of MP-MAB [17] and MST [18]. In general, as long as the expected reward in a problem satisfies the monotonicity and submodularity on the action set, the greedy algorithm serves as an offline oracle to provide an approximate solution [23].

In this paper, we first formulate the CMAB problems with greedy oracle, which is general enough and covers PMC, MP-MAB, and many other problems. In this framework, the objective of the learning agent is to minimize the cumulative greedy regret, defined as the cumulative difference between the expected reward of the selected action and that of the greedy's solution in the real environment. Focusing on a specific PMC problem instance, we derive the hardness analysis of the TS algorithm with the greedy oracle to solve CMAB problems. Due to the mismatch between the estimation gaps that need to be eliminated by exploration and the actual regret the algorithm needs to pay for each such exploration, the TS algorithm with greedy oracle in this CMAB problem cannot achieve as good theoretical performance as previous MAB algorithms. A problem-dependent regret lower bound of order $\Omega(\log T/\Delta^2)$ is provided to illustrate such hardness, where $T$ is the time horizon and $\Delta$ is some reward gap. By carefully exploiting the property of the greedy oracle, we also provide an almost matching problem-dependent regret upper bound, which is tight on that PMC problem instance and also recovers the main order of TS when solving MAB problems [3]. These results are the first theoretical results of TS with approximation oracle to solve CMAB problems, which show that the linear regret example in [30] does not hold for every approximation oracle.

## 2  Related Work

CMAB problems have been widely studied in the literature [11, 6, 7, 29, 30, 24, 32, 21, 14]. Here we mainly focus on the most relevant works. Gai *et al.* [11] first study the CMAB problem with linear reward, where the reward of an action is linear in the reward of base arms included in it. They introduce a UCB-type algorithm to solve such problems and allow approximation algorithms to

serve as offline oracles. Later, Chen *et al.* [6, 7, 29] generalize this framework by considering a larger class of rewards and the case with probabilistically triggered arms (CMAB-T). This framework only assumes the expected reward satisfies the monotonicity and Lipschitz condition on the mean vector of base arms. The combinatorial UCB (CUCB) algorithm is proposed to solve such general CMAB problems, which works with any offline oracle with approximation guarantees. When only approximation oracles are available, the goal of the algorithm is relaxed to minimize the cumulative approximation regret, which is defined as the difference between the expected reward of the selected action and that of the scaled optimal solution. The CUCB algorithm achieves the regret upper bound of order $O(\log T/\Delta_{\min})$ [29], where $\Delta_{\min}$ is the minimum reward gap from the scaled optimal solution over all suboptimal actions.

Compared with UCB-type algorithms which need to compute upper confidence bounds for unknown means of base arms [6, 7, 29], TS-type algorithms do not require the reward function to satisfy the monotonicity on the mean vector of base arms. Benefited from this and other advantages of easier implementation and better practical performances, TS-type algorithms have recently attracted a lot of interest in CMAB problems. Komiyama *et al.* [17] consider using the TS algorithm to solve the MP-MAB problem, where the agent needs to select $K$ from $m$ arms to maximize the sum of rewards over these selected $K$ arms. They provide an optimal regret upper bound of order $O(\log T/\Delta_{K,K+1})$ for the TS algorithm to solve this problem, where $\Delta_{K,K+1}$ is the reward gap between the $K$-th and $(K+1)$-th optimal arm. Later, Wang and Chen [30] consider using TS to solve more general CMAB problems where only the Lipschitz condition is assumed to be satisfied. Their regret upper bound of order $O(\log T/\Delta_{\min})$ matches the main order of the CUCB [29] in the same setting. The coefficient of this upper bound was recently improved by Perrault *et al.* [24], who study the same CMAB setting. Huyuk *et al.* [14] extend the analysis of [30] and consider using the TS algorithm to solve the problem of CMAB-T. However, the current regret upper bound is $O(1/p^*)$ worse than CUCB [29], where $p^*$ is the minimum triggering probability.

All of the above TS-based works need an exact oracle to provide the optimal solution with sampled parameters in each round. However, the exact oracles are usually not feasible as many combinatorial optimization problems, such as the offline problem of OIM and PMC, are NP-hard [16]. Wang and Chen [30] have constructed a problem instance and designed an approximation offline oracle for this problem instance. The analysis has shown that the TS algorithm suffers the linear regret of order $O(T)$ when working with such an approximation oracle. However, this oracle is uncommon and artificial, thus cannot represent the performance of TS when working with common approximation oracles. It is still a significant open problem that whether TS can perform well with approximation oracles.

The greedy algorithm is one of the most important methods with approximation guarantees to solve combinatorial optimization problems. When the mean vector is known beforehand, we call the problem of finding the action with the best expected reward as *offline* problem. Using the greedy algorithm to solve offline combinatorial problems has been studied for decades, including the problem of shortest spanning subtree [18], shortest connection networks [25], set coverage [8], influence maximization [16], and general submodular optimization [23]. There is also a line of studies considering using greedy to solve online problems [4, 19, 10, 27, 22], some of them require the exact reward function forms as prior knowledge. Among these works the most related to ours is [22], both aiming to solve a general class of online problems. Lin *et al.* [22] consider using the online greedy strategy to make decisions based on UCB-type estimators. The algorithm sequentially selects a unit to maximize the current expected reward until no feasible unit can be selected in each round. In their setting, a unit conditioned on a set of previously selected units is regarded as an arm and the marginal reward of selecting this unit is the expected reward of this arm. Since the number of combinations of units is usually exponentially large, there is an exponential number of arms to explore, making the algorithm pay exponential memory cost in practical applications. Based on this framework, the algorithm needs to observe the marginal reward after the decision of each step to update the estimate on the arm. However, such observation may be not available as many combinatorial problems treat the action composed of several units as a whole and select them together. Compared to this work, our framework only needs polynomial memory cost and does not require the observation of the marginal reward.

In this paper, we study the problem of CMAB with the common (approximation) greedy oracle and hope to answer the question of whether the TS algorithm can work well in this setting.

## 3 Setting

The combinatorial multi-armed bandit (CMAB) problem is formulated by a $T$-round learning game between the learning agent and the environment. The environment contains $m$ base arms and the arm set is denoted by $[m] = \{1, 2, \ldots, m\}$. Each arm $i \in [m]$ is associated with a distribution $D_i$ on $[0, 1]$. We consider a combinatorial setting where the agent can select several base arms at a time. In many applications, different arms usually have structural correlations in the selection decision of the agent. For example, in the PMC problem, base arms (edges) starting from the same node must be selected together. Thus the base arm set $[m]$ can be further divided into $n$ units, with each unit containing several base arms and a unit of arms will be selected together. Let $\mathcal{U}$ be the collection of all units and $|s|$ be the number of base arms contained in unit $s$ for any $s \in \mathcal{U}$.

In each round $t = 1, 2, \ldots$, the learning agent selects an action $S_t \in \mathcal{S} = \{S \subseteq \mathcal{U} : |S| = K\}$ to play. Here $\mathcal{S}$ is the set of all candidate actions containing $K$ units. For any action $S$, denote $\cup S = \{i \in s \text{ for some } s \in S\}$ as the set of base arms that belong to units contained in $S$. The environment then draws a random output of all base arms $X_t = (X_{t,1}, X_{t,2}, \ldots, X_{t,m})$ from the distribution $D = D_1 \times D_2 \times \ldots \times D_m$. For any $t$, $X_{t,i}$ is independent and identically distributed on $D_i$ with expectation $\mu_i$, for any base arm $i$. Let $\mu = (\mu_i)_{i \in [m]}$ be the mean vector. We study the semi-bandit feedback [30, 7, 29] where the agent can observe feedback $Q_t = \{(i, X_{t,i}) \mid i \in \cup S_t\}$, namely the output of all base arms in units contained in $S_t$. Denote $\mathcal{H}_t = \{(S_\tau, Q_\tau) : 1 \le \tau < t\}$ as the history of observations at time $t$. The agent finally obtains a corresponding reward $R_t = R(S_t, X_t)$ in this round, which is a function of action $S_t$ and output $X_t$. We assume the expected reward satisfies the following two assumptions, which are standard in CMAB works [7, 30, 14, 24, 29].

**Assumption 1.** *The expected reward of an action $S$ only depends on $S$ and the mean vector $\mu$. That is to say, there exists a function $r$ such that $\mathbb{E}[R_t] = \mathbb{E}_{X_t \sim D}[R(S_t, X_t)] = r(S_t, \mu)$.*

**Assumption 2.** *(Lipschitz continuity) There exists a constant $B$ such that for any action $S$ and mean vectors $\mu, \mu'$, the reward of $S$ under $\mu$ and $\mu'$ satisfies*

$$|r(S, \mu) - r(S, \mu')| \le B \sum_{i \in \cup S} |\mu_i - \mu_i'| . \tag{1}$$

When the mean vector $\mu$ is known beforehand, finding the optimal action containing $K$ units is called the *offline* problem. However, the offline problems are usually NP-hard and enumerating all actions to find the best one is not feasible as the number of actions is exponentially large. The `Greedy` algorithm (presented in Algorithm 1) is a common method to solve such offline problems, which is simple to implement and can provide approximate solutions for OIM [16] and PMC [8], and exact solutions for MP-MAB[17]. More specifically, as long as the reward function satisfies monotonicity and submodularity on the action set, the `Greedy` algorithm can provide solutions with approximate guarantees [23]. Moreover, the `Greedy` algorithm is also popular to serve as a heuristic method in real applications and has good practical performance even without a theoretical guarantee.

---

**Algorithm 1** `Greedy` algorithm
---

1: Input: base arm set $[m]$ and mean vector $\mu = (\mu_i)_{i \in [m]}$, unit set $\mathcal{U}$, action size $K$
2: Initialize: $S_g = \emptyset$
3: **for** $k = 1, 2, \cdots, K$ **do**
4:      $s_k = \mathrm{argmax}_{s \in \mathcal{U} \setminus S_g} r(S_g \cup \{s\}, \mu)$
5:      $S_g = S_g \cup \{s_k\}$
6: **end for**
7: Output: $S_g$

---

We mainly study the CMAB problem with the `Greedy` oracle. With input $\mu = (\mu_i)_{i \in [m]}$, it sequentially selects $K$ units to maximize the current expected reward. To simplify, we assume the `Greedy`'s solution $S_g(\mu)$, abbreviated as $S_g$, is unique, or equivalently the optimal unit in each step $k$ (Line 4 in Algorithm 1) is unique. The general case with multiple solutions can also be solved and would be discussed later. The objective of the learning agent is to maximize the cumulative expected reward over $T$ rounds, or equivalently to minimizing the cumulative expected regret with respect to the

Greedy's solution $S_g$, which we call cumulative *greedy regret* [22] defined by

$$R_g(T) = \mathbb{E}\left[\sum_{t=1}^T \max\{r(S_g, \mu) - r(S_t, \mu), 0\}\right],\tag{2}$$

where the expectation is taken from the randomness in observations and the online algorithm.

We call Greedy an $\alpha$-approximate oracle if $r(S_g(\mu'), \mu') \geq \alpha \cdot r(S^*(\mu'), \mu')$ for any input $\mu'$, where $S^*(\mu')$ is the optimal action under $\mu'$. Note when Greedy is $\alpha$-approximate, the upper bound for greedy regret also implies the upper bound for the $\alpha$-approximate regret defined by the cumulative distance between scaled optimal reward $\alpha \cdot r(S^*(\mu), \mu)$ and $r(S_t, \mu)$ over $T$ rounds. The approximate regret is adopted in previous CMAB works based on UCB-type algorithms [7, 29, 32, 21]. It is much weaker than greedy regret as it relaxes the requirements for online algorithms and only needs them to return solutions satisfying the relaxed approximation ratio. We discuss more on challenges in analyzing the $\alpha$-approximate regret with TS-type algorithms in Section 6.1.

**An example of CMAB: probabilistic maximum coverage (PMC)** The input for the PMC problem is a weighted bipartite graph $G = (L, R, E)$, where each edge $(u, v) \in E$ is associated with a weight $\mu_{(u,v)}$. Denote $\mu = (\mu_{(u,v)})_{(u,v) \in E}$ as the edge weight vector. The goal is to find a node set $S \subseteq L$ with $|S| = K$ to maximize the number of influenced nodes in $R$, where each node $v \in R$ can be influenced by $u \in S$ with independent probability $\mu_{(u,v)}$. The advertisement placement problem can be modeled by PMC, where $L$ is the web page set, $R$ is the user set and $\mu_{(u,v)}$ represents the probability that user $v$ clicks the advertisement on web page $u$. In this application, the user click probabilities are unknown and need to be learned during iterative interactions. The PMC problem fits our CMAB framework with each edge being a base arm and edges starting from the same node forming a unit. The expected reward of an action $S$ is the expected number of nodes finally influenced by it, which is defined as

$$r(S, \mu) = \sum_{v \in R}\left(1 - \prod_{(u,v) \in E, u \in S}\left(1 - \mu_{(u,v)}\right)\right).\tag{3}$$

It is proved that the reward function satisfies Assumption 2 [7] and the Greedy oracle can provide an approximate solution with approximation ratio $(1 - \frac{1}{e})$ for any input [23].

## 4   Algorithm

In this section, we introduce the combinatorial Thompson sampling (CTS) algorithm with Beta priors and Greedy oracle (presented in Algorithm 2) for CMAB problems.

---
**Algorithm 2** CTS algorithm with Beta priors and Greedy oracle
---
1: Input: base arm set $[m]$, unit set $\mathcal{U}$, action size $K$
2: Initialize: $\forall i \in [m], a_i = b_i = 1$
3: **for** $t = 1, 2, \cdots$ **do**
4:     $\forall i \in [m]$ : Sample $\theta_{t,i} \sim \text{Beta}(a_i, b_i)$. Denote $\theta_t = (\theta_{t,1}, \theta_{t,2}, \cdots, \theta_{t,m})$
5:     Select action $S_t = \text{Greedy}([m], \theta_t, \mathcal{U}, K)$ and receive the observation $Q_t$
6:     //Update
7:     **for** $(i, X_{t,i}) \in Q_t$ **do**
8:         With probability $X_{t,i}$, $Y_{t,i} = 1$; with probability $1 - X_{t,i}$, $Y_{t,i} = 0$
9:         Update $a_i = a_i + Y_{t,i}, b_i = b_i + (1 - Y_{t,i})$
10:    **end for**
11: **end for**
---

The algorithm maintains a Beta distribution with parameters $a_i$ and $b_i$ for each base arm $i \in [m]$. In the beginning, it initializes $a_i = b_i = 1, \forall i \in [m]$ (Line 2). In each round $t$, the algorithm first samples a parameter candidate $\theta_{t,i}$ from $\text{Beta}(a_i, b_i)$ representing the current estimate for $\mu_i$ (Line 4). Then the Greedy oracle outputs the solution $S_t$ according to the input vector $\theta_t = (\theta_{t,1}, \theta_{t,2}, \cdots, \theta_{t,m})$ (Line 5). Based on the observation feedback, the algorithm then updates the corresponding Beta distributions for observed base arms (Line 7-10).

# 5 Lower Bound

We investigate the hardness of the CTS algorithm to solve CMAB problems with `Greedy` oracle by proving a problem-dependent regret lower bound.

First, we introduce some notations that will be used in the regret analysis. Recall $S_g$ is the solution returned by the `Greedy` oracle when the input is $\mu$. We denote it as $S_g = \{s_{g,1}, s_{g,2}, \ldots, s_{g,K}\}$, where $s_{g,k}$ is the $k$-th selected unit by `Greedy`. Further, define $S_{g,k} = \{s_{g,1}, s_{g,2}, \ldots, s_{g,k}\}$ as the sequence containing the first $k$ units for any $k \in [K]$. Similarly, let $S_t = \{s_{t,1}, s_{t,2}, \ldots, s_{t,K}\}$ and $S_{t,k} = \{s_{t,1}, s_{t,2}, \ldots, s_{t,k}\}$. Note $S_{g,0} = S_{t,0} = \emptyset$. The corresponding gaps are defined to measure the hardness of the task and the performance of the algorithm.

**Definition 1.** *(Gaps) For any unit $s \in \mathcal{U}$ and index $k \in [K]$ such that $s \notin S_{g,k-1}$, define the marginal reward gap*

$$\Delta_{s,k} = r(S_{g,k}, \mu) - r(S_{g,k-1} \cup \{s\}, \mu)$$

*as the reward difference between $S_{g,k}$ and $S_{g,k-1} \cup \{s\}$. According to the `Greedy` algorithm, we have $\Delta_{s,k} > 0$ for any $k$ such that $s \notin S_{g,k}$. And for any action $S \in \mathcal{S}$, define $\Delta_S = \max\{r(S_g, \mu) - r(S, \mu), 0\}$ as the reward difference from the `Greedy`'s solution $S_g$. Let*

$$\Delta_s^{\min} = \min_{S \in \mathcal{S}: s \in S} \Delta_S, \quad \Delta_s^{\max} = \max_{S \in \mathcal{S}: s \in S} \Delta_S$$

*be the minimum and maximum reward gap of actions containing unit $s$, respectively. Denote $\Delta_{\max} = \max_{S \in \mathcal{S}} \Delta_S$ as the maximum reward gap over all suboptimal actions.*

We take the following PMC problem (shown in Figure 1) as the instance to carry out the hardness analysis. Each edge in the graph is a base arm and the set of all outgoing edges from a single node forms a unit. The action size is set to $K = 2$. The weight $\mu_{(u,v)}$ of each edge $(u, v)$ is listed on the edges, where we assume $0 < \Delta \leq 0.04$. The expected reward $r(S, \mu)$ of an action $S$ under $\mu$ is defined as Eq (3). For example, when $u_1$ and $u_2$ are selected, the probability of $v_1$ being influenced is $1 - (1 - \mu_{(u_1,v_1)})(1 - \mu_{(u_2,v_1)}) = 0.592$ and the probability that $v_2$ is influenced is $\mu_{(u_2,v_2)} = 0.3$. The expected reward of $S = \{u_1, u_2\}$ is $r(S, \mu) = 0.592 + 0.3 = 0.892$. For simplicity, we also assume the output of each base arm in each round is exactly its mean. This assumption still satisfies the above properties and is also adopted in previous lower bound proofs [3] to simplify the analysis.

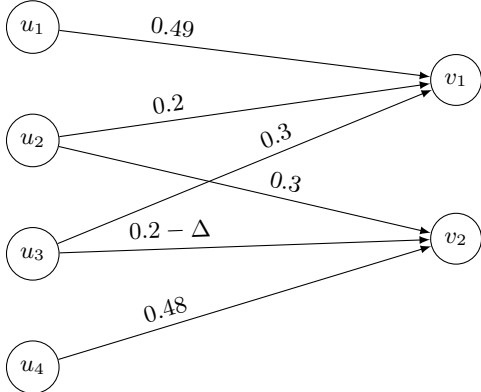

Figure 1: The underlying graph of the PMC instance used to derive the hardness analysis.

For convenience, we first list the expected reward of each action in this problem in Table 1. We can find that the greedy solution is $S_g = \{u_2, u_1\}$ with $s_{g,1} = u_2$, $s_{g,2} = u_1$, and $r(S_g, \mu) = 0.892$, while the optimal action is $\{u_1, u_4\}$. The corresponding marginal reward gaps of each unit can be then computed as follows.

$$\Delta_{u_1,1} = 0.01\,;$$
$$\Delta_{u_3,1} = \Delta, \qquad \Delta_{u_3,2} = 0.012 + 0.7\Delta\,;$$
$$\Delta_{u_4,1} = 0.02, \quad \Delta_{u_4,2} = 0.056\,.$$

| Action | Expected Reward | Action | Expected Reward |
|--------|-----------------|--------|-----------------|
| $\{u_1\}$ | 0.49 | $\{u_1, u_2\}$ | 0.892 |
| $\{u_2\}$ | 0.5 | $\{u_1, u_3\}$ | $0.843 - \Delta$ |
| $\{u_3\}$ | $0.5 - \Delta$ | $\{u_1, u_4\}$ | 0.97 |
| $\{u_4\}$ | 0.48 | $\{u_2, u_3\}$ | $0.88 - 0.7\Delta$ |
| $\{u_3, u_4\}$ | $0.884 - 0.52\Delta$ | $\{u_2, u_4\}$ | 0.836 |

Table 1: The expected rewards of actions in the problem instance shown in Figure 1.

In the following, we mainly focus on unit $u_3$ and take it as an example to derive the hardness analysis. According to Table 1, all actions containing $u_3$ are suboptimal actions compared to $S_g$ and $\Delta_{u_3}^{\min} = \Delta_{\{u_3, u_4\}} = 0.52\Delta + 0.008$. Thus to avoid regret generation, the algorithm should avoid incorporating $u_3$ in the action $S_t$. Intuitively, $u_3$ should be explored at least $\Omega\left(\log T/\Delta_{u_3,1}^2\right) = \Omega\left(\log T/\Delta^2\right)$ times to be distinguished from $s_{g,1} = u_2$ and thus can avoid being selected as the first unit by Greedy. However, in each round of exploration for $u_3$, the algorithm needs to pay a constant regret of at least $0.52\Delta + 0.008$. Thus the estimation gap $\Delta$ needs to be eliminated by exploration on the denominator of $\Omega\left(\log T/\Delta^2\right)$ cannot be canceled by the actual regret paid in each exploration round. Such mismatch would cause the greedy regret at least of order $\Omega\left(\log T/\Delta^2\right)$.

We give the formal lower bound for both the expected number of selections of each unit and the cumulative greedy regret in the following Theorem 1.

**Theorem 1.** *(Lower bound) Using the CTS algorithm with Gaussian priors and Greedy oracle to solve the CMAB problem shown in Figure 1, when $T$ is sufficiently large, we have*

$$\mathbb{E}\left[N_{T+1,s}\right] = \Omega\left(\frac{\log T}{\Delta_{s,1}^2}\right), \tag{4}$$

*for any $s \neq s_{g,1} = u_2$, where $N_{T+1,s} = \sum_{t=1}^{T} \mathbb{1}\{s \in S_t\}$ is the number of rounds when $s$ is contained in the selected action set $S_t$.*

*Further, the cumulative greedy regret satisfies*

$$R_g(T) = \Omega\left(\frac{\log T}{\Delta_{u_3,1}^2}\right) = \Omega\left(\frac{\log T}{\Delta^2}\right). \tag{5}$$

The proof of Theorem 1 follows directly the intuition of the above hardness analysis. Due to the space limit, we include the detailed proof in Appendix B. The reason why we consider using Gaussian priors to derive the lower bound analysis is that we hope to use its concentration and anti-concentration inequalities. The analysis can directly apply to other types of prior distributions if similar inequalities can be provided. The main operations of CTS with Gaussian priors are very similar to that of Algorithm 2, while the only difference is on the prior distribution for unknown parameters and the corresponding updating mechanism. To be self-contained, we also present the detailed CTS algorithm with Gaussian priors in Appendix B.

Lin *et al.* [22] also show a lower bound for greedy regret of order $\Omega(\log T/\Delta^2)$ with $\Delta \in (0, 1/4)$. However, the problem instance used to derive this lower bound is not a CMAB problem and thus their result is not comparable with Theorem 1.

## 6 Upper Bound

By investigating the properties of the CTS algorithm and the Greedy oracle, we also provide a problem-dependent regret upper bound for Algorithm 2 to solve general CMAB problems.

**Theorem 2.** *(Upper bound) The cumulative greedy regret of Algorithm 2 can be upper bounded by*

$$R_g(T) \leq \sum_{s \neq s_{g,1}} \max_{k:s \notin S_{g,k}} \frac{6B^2 |s|^2 \Delta_s^{\max} \log T}{\left(\Delta_{s,k} - 2B |\cup S_g| \varepsilon\right)^2} + \sum_{k \in [K]} \frac{C}{\varepsilon^2} \left(\frac{C'}{\varepsilon^4}\right)^{|s_{g,k}|} \Delta_{\max}$$

$$+ \left( |\cup S_g| \left( 2 + \frac{8}{\varepsilon^2} \right) + 4m \right) \Delta_{\max} \tag{6}$$

$$= O \left( \sum_{s \neq s_{g,1}} \max_{k:s \notin S_{g,k}} \frac{B^2 |s|^2 \Delta_{\max} \log T}{\Delta_{s,k}^2} \right), \tag{7}$$

*for any $\varepsilon$ such that $\forall s \neq s_{g,1}$ and $k$ satisfying $s \notin S_{g,k}$, $\Delta_{s,k} > 2B |\cup S_g| \varepsilon$, where $B$ is the coefficient of the Lipschitz continuity condition, $|\cup S_g|$ is the number of base arms that belong to the units contained in $S_g$, $C$ and $C'$ are two universal constants.*

Due to the space limit, we provide the proof sketch of Theorem 2 in Section 6.2 and defer the formal proof to Appendix C. In order to better compare the upper and lower bounds, we also analyze the greedy regret of the CTS algorithm with Gaussian priors in Appendix D, which achieves the same order of the upper bound with Theorem 2 only up to some constant factors.

## 6.1 Discussions

**Challenges in analyzing the $\alpha$-approximate regret with CTS** The $\alpha$-approximate regret is first brought up in analyzing UCB-type algorithms [6, 7, 29]. Under UCB, benefiting from the monotonicity between the true parameter $\mu$ and the UCB parameter $\bar{\mu}$, the $\alpha$-approximate regret can be deducted as

$$\alpha \cdot r(S^*, \mu) - r(S_t, \mu) \leq \alpha \cdot r(S^*, \bar{\mu}) - r(S_t, \mu) \leq r(S_t, \bar{\mu}) - r(S_t, \mu) \leq \sum_{i \in \cup S_t} |\bar{\mu}_i - \mu_i|,$$

where $S^* \in \arg\max_{S \in \mathcal{S}} r(S, \mu)$ is an exact optimal action under real parameter $\mu$. Thus it only needs to bound the number of selections of bad action $S_t$ to get an upper bound for the $\alpha$-approximate regret. However, under CTS, since there is no monotonicity between the true parameter $\mu$ and the surrogate parameter $\theta$, the approximate regret can only be deducted as

$$\begin{aligned} \alpha \cdot r(S^*, \mu) - r(S_t, \mu) &\leq \alpha \cdot r(S^*, \mu) - \alpha \cdot r(S^*, \theta) + \alpha \cdot r(S^*, \theta) - r(S_t, \mu) \\ &\leq \alpha \cdot r(S^*, \mu) - \alpha \cdot r(S^*, \theta) + r(S_t, \theta) - r(S_t, \mu) \\ &\leq \alpha \sum_{i \in \cup S^*} |\theta_i - \mu_i| + \sum_{i \in \cup S_t} |\theta_i - \mu_i|. \end{aligned}$$

To get an upper bound for the RHS, it requires a sufficient number of selections of the exact optimal action $S^*$, which may not be the case with approximate oracles like the example shown in Theorem 1. Thus the $\alpha$-approximate regret may not well fit TS-type algorithms.

**Tightness of the upper bound** We now discuss the tightness of the regret upper bound in Theorem 2 based on the problem instance shown in Figure 1. Specific to this problem, we have $|s| \leq 2$ for all $s$ since each node has no more than 2 outgoing edges. And based on [21, Theorem 4], the coefficient of the Lipschitz condition in this problem is $B = 1$. When $0 < \Delta \leq 0.04$, we have $\Delta_{s,1} = \min_{k:s \notin S_{g,k}} \Delta_{s,k}$ for any $s \neq s_{g,1} = u_2$, and $\Delta_{\max} = \Delta_{\{u_2,u_4\}} = 0.056$ is a constant. Thus our regret upper bound in this problem instance is of order

$$O \left( \sum_{s \neq s_{g,1}} \max_{k:s \notin S_{g,k}} \frac{B^2 |s|^2 \Delta_{\max} \log T}{\Delta_{s,k}^2} \right) = O \left( \sum_{s \in \{u_1,u_3,u_4\}} \frac{\log T}{\Delta_{s,1}^2} \right) = O \left( \frac{\log T}{\Delta^2} \right),$$

where the last equality holds since $\Delta_{u_1,1} = 0.01$, $\Delta_{u_4,1} = 0.02$ are constants and $\Delta_{u_3,1} = \Delta$.

We can see our regret upper bound matches the lower bound of (5) in Theorem 1 only up to some constant factors in this specific problem instance.

**Comparison with MAB** When each unit contains only one base arm and the action size is $K = 1$, our CMAB framework recovers the MAB problem and the `Greedy` oracle can provide the exact optimal solution. Thus we can also compare our regret upper bound with the theoretical results of the TS algorithm in MAB problems. In the MAB problem, the expected reward of each action is exactly the mean of the base arm contained in this action. Thus the Lipschitz coefficient is just $B = 1$ and $|s| = 1$ for any unit $s$. The optimal action is $S_g = S_{g,1}$ with $|\cup S_g| = 1$. And for any unit $s \neq s_{g,1}$,

we have $\Delta_s^{\max} = \Delta_{s,1}$. Thus, according to (6) of Theorem 2, the regret upper bound of Algorithm 2 in MAB problems is of order $O\left(\sum_{s \neq s_{g,1}} \frac{\log T}{\Delta_{s,1}}\right)$, which recovers the main order of the regret upper bound of TS for MAB problems [3].

**Comparison with [22]**    Though Lin *et al.* [22] also study greedy regret, the results are not directly comparable in general since the setting studied in [22] is not a CMAB setting. We find that the PMC problem under a bar graph in these two settings can be equivalent, where a bar graph is a special bipartite graph with each left node's outdegree being 1 (indegree being 0) and each right node's indegree being 1 (outdegree being 0). In this case, our greedy regret upper bound is of order $O(m \log T / \Delta^2)$ and theirs is $O(mK \log T / \Delta^2)$. So ours is $O(K)$ better than theirs. Even in this case, their algorithm needs to estimate $O(m \cdot 2^m)$ parameters, while Algorithm 2 is more efficient and only needs to estimate $O(m)$ parameters.

**The definition of the marginal reward gap**    Recall that the Greedy oracle provides approximate solutions for problems whose expected reward satisfies monotonicity and submodularity on the action set. Formally, the submodularity means for any action $S \subseteq T$ and unit $s \notin T$, there is $r(S \cup \{s\}, \mu) - r(S, \mu) \geq r(T \cup \{s\}, \mu) - r(T, \mu)$, which characterizes the phenomenon of diminishing returns. One may concern that in these problems, due to the submodularity, the marginal reward gap $\Delta_{s,k}$ for larger $k$ may become much smaller and the main order of the upper bound thus blows up. We clarify that the submodularity cannot imply the relationships among marginal reward gaps $\Delta_{s,k}$ for different $k \in [K]$. The problem instance in Figure 1 satisfying submodularity [16] indicates that the marginal reward gap $\Delta_{s,k}$ does not necessarily decrease with the increase of $k$.

## 6.2   Proof Sketch

In this section, we briefly introduce the proof idea of Theorem 2. At any step $k \in [K]$, for any unit $s \notin S_{g,k}$, the selection of $s$ in action $S_t$ may force the Greedy oracle to choose a worse action in subsequential steps and make CTS suffer constant regret. A sufficient condition for generating zero greedy regret in round $t$ is that each unit $s_{t,k}$ selected in step $k$ is actually $s_{g,k}$ for any $k \in [K]$. Thus to bound the cumulative greedy regret, we sequentially analyze whether $s_{g,k}$ is appropriately selected at each step $k$.

Recall that under the framework of the CTS algorithm, the Greedy oracle sequentially selects $s_{t,k}$ for $k \in [K]$ based on $\theta_t$ sampled from posterior distributions in round $t$. Focusing on step $k = 1$, to ensure $s_{t,1} = s_{g,1}$, the algorithm needs to guarantee the accurate estimations $\theta_{t,s}$ for any unit $s \in \mathcal{U}$ such that $r(\{s\}, \theta_t) < r(S_{g,1}, \theta_t), \forall s \neq s_{g,1}$, where $\theta_{t,s}$ is the projection of $\theta_t$ on unit $s$. We first assume $s_{g,1}$ is already estimated well. Then based on this assumption, when all of the other units $s \neq s_{g,1}$ have been explored $O(\log T / \Delta_{s,1}^2)$ times and thus estimated accurately, $s_{g,1}$ would be selected at the first step with high probability. But if after any other unit $s \neq s_{g,1}$ has already been estimated well, $s_{g,1}$ is still not selected appropriately, we can conclude that the estimations for $s_{g,1}$ are not accurate enough. In this case, the Beta posterior for $s_{g,1}$ tends to be uniformly distributed. When CTS sample $\theta_{t,s_{g,1}}$ from its Beta posterior, with constant probability there would be $r(S_{g,1}, \theta_t) > r(\{s\}, \mu) \approx r(\{s\}, \theta_t)$ for any unit $s \neq s_{g,1}$. Thus after some rounds, $s_{g,1}$ would be selected for enough times and also estimated accurately. In the following rounds, $s_{g,1}$ would be selected appropriately at the first step with high probability. Above all, the expected number of misselections at the first step can be bounded.

The above analysis can apply to other cases when $k = 2, 3, \ldots, K$. Based on the correct selections in the first $k - 1$ steps, the misselection of $s_{t,k}$ also comes from the bad estimations for both $s_{g,k}$ and other units $s \notin S_{g,k}$. To distinguish $r(S_{g,k-1} \cup \{s\}, \theta_t)$ from $r(S_{g,k}, \theta_t)$, those units need to be explored at least $O(\log T / \Delta_{s,k}^2)$ times.

According to the above analysis, for each unit $s \neq s_{g,1}$, we define the exploration price as

$$L(s) = O\left(\max_{k:s \notin S_{g,k}} \frac{\log T}{\Delta_{s,k}^2}\right). \tag{8}$$

To avoid being incorrectly selected at some step $k$, each unit $s \notin S_{g,k}$ needs to be explored for at least $L(s)$ times. The sum of $L(s)$ over all units $s \neq s_{g,1}$ leads to the main order of the regret upper bound in Theorem 2.

### 6.3 Extension to Multiple-solution Case

We can also extend the analysis of the regret upper bound to the case where multiple solutions may be returned by Greedy with input $\mu$, or equivalently the optimal unit in each step $k$ (Line 4 in Algorithm 1) may not be unique. Let

$$\sigma_K = \left\{ \{s_1, s_2, \ldots, s_K\} : s_1 \in \operatorname*{argmax}_s r(\{s\}, \mu), \ldots, s_K \in \operatorname*{argmax}_{s \notin \{s_1, \ldots, s_{K-1}\}} r(\{s_1, \ldots, s\}, \mu) \right\}$$

be the set of all actions that are possibly returned by the Greedy oracle when the input is $\mu$. Here we do not care how Greedy breaks the tie at each step and regard this process as a black box. In order to take into account the worst case where Greedy always returns a solution with minimum reward compared to other possible solutions, we define $S_g \in \operatorname*{argmin}_{S \in \sigma_K} r(S, \mu)$ as one of Greedy's possible solutions with the minimum expected reward and consider the cumulative greedy regret defined in Eq (2).

The regret analysis of Algorithm 2 in this case is similar to the proof of Theorem 2. A sufficient condition for generating zero regret in round $t$ is that the selected action $S_t$ falls into the set $\sigma_K$. Thus to bound the cumulative greedy regret, we sequentially analyze whether the unit $s_{t,k}$ selected in each step $k$ is an optimal unit conditioned on the previously selected units $S_{t,k-1}$. For completeness, we include the regret upper bound as well as the detailed proof for this case in Appendix E.

## 7 Conclusion

In this paper, we aim to answer the question of whether the convergence analysis of TS can be extended beyond exact oracles in the CMAB area. Taking the common offline (approximation) Greedy oracle as an example, we derive the hardness analysis of CTS for CMAB problems based on a constructed CMAB problem instance. When using CTS with Greedy oracle to solve this problem, we find that the algorithm needs to explore at least $\Omega(\log T/\Delta^2)$ rounds to distinguish suboptimal units from the optimal unit at some step. However, at least constant regret needs to be paid for each exploration round. The mismatch between the gap to be distinguished and the actually paid regret forces the algorithm to pay the cumulative greedy regret of order $\Omega(\log T/\Delta^2)$. We also provide an almost matching problem-dependent regret upper bound for the CTS algorithm with Greedy oracle to solve CMAB problems. The upper bound is tight on the constructed problem instance only up to some constant factors and also recovers the main order of TS when solving MAB problems.

An interesting future direction is to extend the current CMAB framework to the case with probabilistically triggered arms (CMAB-T). The CMAB-T framework can model the OIM problem on general social networks. As shown in [21], using UCB-type algorithms to solve such a problem may face great challenges on the computation efficiency. This problem is expected to be avoided by TS-type algorithms since TS would sample candidate parameters to escape the computation of complicated optimization problems. However, the current proof idea based on each selection step of the Greedy oracle (proof of Lemma 1) cannot directly apply to this setting as different units may probabilistically trigger some common base arms. New proof techniques are required to derive the theoretical guarantee of the TS algorithm with the Greedy oracle in this framework.

## Acknowledgement

The corresponding author Shuai Li is supported by National Natural Science Foundation of China (62006151, 62076161). This work is sponsored by Shanghai Sailing Program.

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
