## A  Notations

Before the proofs, we first introduce some notations that would be used in the regret analysis.

For any base arm $i \in [m]$, let $N_{t,i} = \sum_{\tau < t} \mathbb{1}\{i \in \cup S_\tau\}$ be the number of observations of $i$ and $\hat{\mu}_{t,i} = \frac{1}{N_{t,i}} \sum_{\tau < t : i \in \cup S_\tau} X_{\tau,i}$ be the empirical mean outcome of $i$ before the start of round $t$. Denote $\hat{\mu} = (\hat{\mu}_{t,1}, \hat{\mu}_{t,2}, \ldots, \hat{\mu}_{t,m})$ as the empirical mean vector. Since a unit of base arms will be selected together, we abuse a bit the notation $N_{t,s} = \sum_{\tau < t} \mathbb{1}\{s \in S_\tau\}$ to represent the number of selections of unit $s$ before the start of round $t$. For any mean vector $\theta \in [0,1]^m$, denote $\theta_s$ and $\theta_S$ as the projection of $\theta$ on unit $s$ and action $S$.

## B  Proof of Theorem 1

To be self-contained, we present the detailed CTS algorithm with Gaussian priors and `Greedy` oracle in Algorithm 3. The main operation of the algorithm is the same as Algorithm 2, the only difference is on the prior distribution for unknown parameters and the corresponding updating mechanism. As we previously discussed, the reason for using the Gaussian distribution to derive the lower bound analysis is that we hope to use its concentration and anti-concentration inequalities. The analysis directly applies to other types of prior distributions if similar inequalities can be provided.

In this algorithm, an initialization phase is introduced to ensure at least one observation has been collected for each base arm (Line 2). In each round $t$, the posterior for $\mu_i$ is given by $\mathcal{N}(\hat{\mu}_{t,i}, \frac{1}{N_{t,i}})$, a Gaussian distribution with mean $\hat{\mu}_{t,i}$ and variance $\frac{1}{N_{t,i}}$. If arm $i$ is observed in this round, the mean and variance of its Gaussian distribution will be updated as Line 7.

---

**Algorithm 3** CTS algorithm with Gaussian priors and `Greedy` oracle

1: Input: base arm set $[m]$, unit set $\mathcal{U}$, action size $K$
2: Initialization: for each unit $s \in \mathcal{U}$, select an arbitrary action $S \in \mathcal{S}$ such that $s \in S$. Update $N_{1,i}$ and $\hat{\mu}_{1,i}$ for any $i \in [m]$ based on observations in this phase.
3: **for** $t = 1, 2, \cdots$ **do**
4: $\quad \forall i \in [m]$ : Sample $\theta_{t,i} \sim \mathcal{N}(\hat{\mu}_{t,i}, \frac{1}{N_{t,i}})$. Denote $\theta_t = (\theta_{t,1}, \theta_{t,2}, \cdots, \theta_{t,m})$
5: $\quad$ Select action $S_t = \text{Greedy}([m], \theta_t, \mathcal{U}, K)$ and receive the observation $Q_t$
6: $\quad$ **for** $(i, X_{t,i}) \in Q_t$ **do**
7: $\quad\quad \hat{\mu}_{t+1,i} = \frac{\hat{\mu}_{t,i} N_{t,i} + X_{t,i}}{N_{t,i} + 1}, N_{t+1,i} = N_{t,i} + 1$
8: $\quad$ **end for**
9: **end for**

---

For any unit $s \neq s_{g,1}$, in any round $t$, define event

$$A_s(t) = \left\{ N_{t,s} \geq \frac{2|s|}{7\Delta_{s,1}^2} \left( \log T + \log \Delta_{s,1}^2 - \log \log T \right) \right\}$$

When $\mathbb{P}(A_s(t)) \geq \frac{1}{2}$, we have

$$\mathbb{E}[N_s(T+1)] \geq \mathbb{E}[N_s(t)] \geq \mathbb{E}[N_s(t) \mid A_s(t)] \cdot \mathbb{P}(A_s(t))$$
$$\geq \frac{1}{2} \cdot \frac{2|s|}{7\Delta_{s,1}^2} \left( \log T + \log \Delta_{s,1}^2 - \log \log T \right)$$
$$= \Omega\left( \frac{\log T}{\Delta_{s,1}^2} \right),$$

when $T$ is sufficiently large.

If $\mathbb{P}(A_s(t)) < \frac{1}{2}$ in round $t$, we then prove unit $s$ will be selected by `Greedy` in the following rounds with a large probability and the expected number of its selections will finally exceed this threshold.

$$\mathbb{P}(s \in S_t) \geq \mathbb{P}(s = s_{t,1})$$

$$\geq \mathbb{P}\left(r(\{s\},\theta_t) > r(S_{g,1},\mu), \forall s' \neq s : r(\{s'\},\theta_t) \leq r(S_{g,1},\mu)\right)$$
$$= \mathbb{P}\left(r(\{s\},\theta_t) > r(S_{g,1},\mu)\right) \cdot \mathbb{P}\left(\forall s' \neq s : r(\{s'\},\theta_t) \leq r(S_{g,1},\mu)\right)$$

where the last equality holds since different units have no common base arms and thus the events on their rewards are independent of that on others.

For the first part, we have

$$\mathbb{P}\left(r(\{s\},\theta_t) > r(S_{g,1},\mu)\right)$$
$$= \mathbb{P}\left(r(\{s\},\theta_t) - r(\{s\},\mu) > r(S_{g,1},\mu) - r(\{s\},\mu)\right)$$
$$= \mathbb{P}\left(r(\{s\},\theta_t) - r(\{s\},\mu) > \Delta_{s,1}\right)$$
$$= \mathbb{P}\left(\sum_{i \in s}(\theta_{t,i} - \mu_i) > \Delta_{s,1}\right)$$
$$\geq \mathbb{P}\left(\sum_{i \in s}(\theta_{t,i} - \mu_i) > \Delta_{s,1} \mid \neg A_s(t)\right)\mathbb{P}\left(\neg A_s(t)\right)$$
$$\geq \frac{1}{2}\mathbb{P}\left(\sum_{i \in s}(\theta_{t,i} - \mu_i)\sqrt{\frac{N_{t,s}}{|s|}} > \Delta_{s,1}\sqrt{\frac{N_{t,s}}{|s|}} \mid \neg A_s(t)\right)$$
$$> \frac{1}{16\sqrt{\pi}}\exp\left(-\frac{7\Delta_{s,1}^2}{2|s|} \times \left(\frac{2|s|}{7\Delta_{s,1}^2}\left(\log T + \log\Delta_{s,1}^2 - \log\log T\right)\right)\right)$$
$$= \frac{1}{16\sqrt{\pi}}\frac{\log T}{T \cdot \Delta_{s,1}^2},$$

where the third equality is due to the reward definition in this specific problem instance and the last inequality is due to the Lemma 6 and the event $\neg A_s(t)$.

For the second part, we have

$$\mathbb{P}\left(\forall s' \neq s : r(\{s'\},\theta_t) \leq r(S_{g,1},\mu)\right)$$
$$= \mathbb{P}\left(r(S_{g,1},\theta_t) \leq r(S_{g,1},\mu)\right) \cdot \mathbb{P}\left(\forall s' \notin \{s,s_{g,1}\} : r(\{s'\},\theta_t) \leq r(S_{g,1},\mu)\right)$$
$$= \mathbb{P}\left(\sum_{i \in s_{g,1}}\theta_{t,i} \leq \sum_{i \in s_{g,1}}\mu_i\right) \cdot \mathbb{P}\left(\forall s' \notin \{s,s_{g,1}\} : r(\{s'\},\theta_t) \leq r(S_{g,1},\mu)\right)$$
$$\geq \mathbb{P}\left(\forall i \in s_{g,1} : \theta_{t,i} \leq \mu_i\right) \cdot \mathbb{P}\left(\forall s' \notin \{s,s_{g,1}\} : r(\{s'\},\theta_t) \leq r(S_{g,1},\mu)\right)$$
$$\geq \left(\frac{1}{2}\right)^{|s_{g,1}|}\mathbb{P}\left(\forall s' \notin \{s,s_{g,1}\} : r(\{s'\},\theta_t) \leq r(S_{g,1},\mu)\right)$$
$$= \left(\frac{1}{2}\right)^{|s_{g,1}|}\prod_{s' \notin \{s,s_{g,1}\}}\mathbb{P}\left(r(\{s'\},\theta_t) \leq r(S_{g,1},\mu)\right),$$

where the first and last equality is again due to the independence over different units, the last inequality comes from the result of Lemma 6 and the independence over base arms. For each term in the last formula, there is

$$\mathbb{P}\left(r(\{s'\},\theta_t) \leq r(S_{g,1},\mu)\right)$$
$$= \mathbb{P}\left(r(\{s'\},\theta_t) - r(\{s'\},\mu) \leq r(S_{g,1},\mu) - r(\{s'\},\mu)\right)$$
$$= \mathbb{P}\left(r(\{s'\},\theta_t) - r(\{s'\},\mu) \leq \Delta_{s',1}\right)$$
$$= \mathbb{P}\left(\sum_{i \in s'}(\theta_{t,i} - \mu_i) \leq \Delta_{s',1}\right)$$
$$= \mathbb{P}\left(\sum_{i \in s'}(\theta_{t,i} - \mu_i)\sqrt{\frac{N_{t,s'}}{|s'|}} \leq \Delta_{s',1}\sqrt{\frac{N_{t,s'}}{|s'|}}\right)$$

$$\geq 1 - \frac{1}{4} \exp\left(-\frac{\Delta_{s',1}^2 N_{t,s'}}{2|s'|}\right)$$

$$\geq 1 - \frac{1}{4} \exp\left(-\frac{\Delta_{s',1}^2}{2|s'|}\right),$$

where the first inequality comes from Lemma 6 and the last one is because $N_{t,s} \geq 1$ after the initialization phase for any $s$. Thus we have

$$\mathbb{P}\left(\forall s' \neq s : r(\{s'\}, \theta(t)) \leq r(S_{g,1}, \mu)\right) \geq \left(\frac{1}{2}\right)^{|s_{g,1}|} \prod_{s' \notin \{s, s_{g,1}\}} \left(1 - \frac{1}{4}\exp\left(-\frac{\Delta_{s',1}^2}{2|s'|}\right)\right) \triangleq C'',$$

here $C''$ can be regarded as a constant. Above all, we have

$$\mathbb{P}\left(s \in S_t\right) \geq C'' \frac{1}{16\sqrt{\pi}} \frac{\log T}{T \cdot \Delta_{s,1}^2} \triangleq p,$$

the total expected number of rounds when unit $s$ is selected in $S_t$ is at least

$$\mathbb{E}\left[N_{T+1,s}\right] \geq Tp = \Omega\left(\frac{\log T}{\Delta_{s,1}^2}\right).$$

Thus we have proved (4) in Theorem 1.

And in the above CMAB instance, we can find that

$$\Delta_{u_3}^{\min} = 0.52\Delta + 0.008 = 0.52\Delta_{u_3,1} + 0.008,$$

Above all,

$$
\begin{aligned}
R_g(T) = \mathbb{E}\left[\sum_{t=1}^T \Delta_{S_t}\right] &\geq \mathbb{E}\left[\sum_{t=1}^T \mathbb{1}\{u_3 \in S_t\}\Delta_{S_t}\right] \\
&\geq \mathbb{E}\left[N_{T+1,u_3}\right]\Delta_{u_3,\min} \\
&= \Omega\left(\frac{\log T}{\Delta_{u_3,1}^2} \cdot (0.52\Delta_{u_3,1} + 0.008)\right) \\
&= \Omega\left(\frac{\log T}{\Delta_{u_3,1}^2}\right).
\end{aligned}
$$

We then complete the proof of (5) in Theorem 1.

## C  Proof of Theorem 2

For any unit $s \neq s_{g,1}$, we define the exploration price $L(s)$ as

$$L(s) = \max_{k:s\notin S_{g,k}} \frac{6B^2 |s|^2 \log T}{(\Delta_{s,k} - 2B|\cup S_g|\varepsilon)^2}.$$

In each round $t$, define the following event

$$B(t) = \left\{\exists i \in [m] : |\theta_{t,i} - \hat{\mu}_{t,i}| > \sqrt{\frac{3\log T}{2N_{t,i}}}\right\}, \quad C(t) = \left\{\exists i \in [m] : |\hat{\mu}_{t,i} - \mu_i| > \sqrt{\frac{3\log T}{2N_{t,i}}}\right\}.$$

The greedy regret then can be decomposed as

$$R_g(T) \leq \mathbb{E}\left[\sum_{t=1}^T \mathbb{1}\{\neg B(t), \neg C(t)\}\Delta_{S_t}\right] + \mathbb{E}\left[\sum_{t=1}^T \mathbb{1}\{B(t)\}\Delta_{S_t}\right] + \mathbb{E}\left[\sum_{t=1}^T \mathbb{1}\{C(t)\}\Delta_{S_t}\right].$$

$$(9)$$

We then bound these three terms in (9) one by one.

**The first term in** (9):

Recall $\Delta_{S_t} = \max\{r(S_g, \mu) - r(S_t, \mu), 0\}$. If $s_{t,k} = s_{g,k}$ for any $k \in [K]$, we must have $\Delta_{S_t} = 0$. Thus to bound this term, we analyze the relationship between $s_{t,k}$ and $s_{g,k}$ sequentially for $k = 1, 2, \ldots, K$. According to this idea, this term can be bounded by

$$\mathbb{E}\left[\sum_{t=1}^T \mathbb{1}\{\neg B(t), \neg C(t)\} \Delta_{S_t}\right]$$

$$\leq \sum_{k \in [K]} \mathbb{E}\left[\sum_{t=1}^T \mathbb{1}\left\{\neg B(t), \neg C(t), S_{t,k-1} = S_{g,k-1}, \left\|\theta_{t,S_{g,k-1}} - \mu_{S_{g,k-1}}\right\|_\infty \leq \varepsilon, s_{t,k} \neq s_{g,k}\right\} \Delta_{S_t}\right]$$

$$+ \sum_{k \in [K]} \mathbb{E}\left[\sum_{t=1}^T \mathbb{1}\left\{s_{t,k} = s_{g,k}, \left\|\theta_{t,s_{g,k}} - \mu_{s_{g,k}}\right\|_\infty > \varepsilon\right\}\right] \cdot \Delta_{\max}.$$

According to Lemma 1, the first term can be bounded by

$$\sum_{k \in [K]} \mathbb{E}\left[\sum_{t=1}^T \mathbb{1}\left\{\neg B(t), \neg C(t), S_{t,k-1} = S_{g,k-1}, \left\|\theta_{t,S_{g,k-1}} - \mu_{S_{g,k-1}}\right\|_\infty \leq \varepsilon, s_{t,k} \neq s_{g,k}\right\} \Delta_{S_t}\right]$$

$$\leq \sum_{k \in [K]} \mathbb{E}\left[\sum_{s \notin S_{g,k}} \sum_{t=1}^T \mathbb{1}\{s = s_{t,k}, N_{t,s} \leq L(s)\} \Delta_{S_t}\right] + \sum_{k \in [K]} \frac{C}{\varepsilon^2}\left(\frac{C'}{\varepsilon^4}\right)^{|s_{g,k}|} \Delta_{\max}$$

$$\leq \mathbb{E}\left[\sum_{s \neq s_{g,1}} \sum_{t=1}^T \sum_{k \in [K]} \mathbb{1}\{s = s_{t,k}, N_{t,s} \leq L(s)\} \Delta_{S_t}\right] + \sum_{k \in [K]} \frac{C}{\varepsilon^2}\left(\frac{C'}{\varepsilon^4}\right)^{|s_{g,k}|} \Delta_{\max}$$

$$\leq \mathbb{E}\left[\sum_{s \neq s_{g,1}} \sum_{t=1}^T \mathbb{1}\{s \in S_t, N_{t,s} \leq L(s)\} \Delta_{S_t}\right] + \sum_{k \in [K]} \frac{C}{\varepsilon^2}\left(\frac{C'}{\varepsilon^4}\right)^{|s_{g,k}|} \Delta_{\max}$$

$$\leq \sum_{s \neq s_{g,1}} L(s) \Delta_s^{\max} + \sum_{k \in [K]} \frac{C}{\varepsilon^2}\left(\frac{C'}{\varepsilon^4}\right)^{|s_{g,k}|} \Delta_{\max}$$

$$\leq \sum_{s \neq s_{g,1}} \max_{k: s \notin S_{g,k}} \frac{6B^2 |s|^2 \Delta_s^{\max} \log T}{\left(\Delta_{s,k} - 2B |\cup S_g| \varepsilon\right)^2} + \sum_{k \in [K]} \frac{C}{\varepsilon^2}\left(\frac{C'}{\varepsilon^4}\right)^{|s_{g,k}|} \Delta_{\max},$$

where $C, C'$ are two universal constants. According to Lemma 3, the second term can be bounded by

$$\sum_{k \in [K]} \mathbb{E}\left[\sum_{t=1}^T \mathbb{1}\left\{s_{t,k} = s_{g,k}, \left\|\theta_{t,s_{g,k}} - \mu_{s_{g,k}}\right\|_\infty > \varepsilon\right\}\right] \Delta_{\max} \leq \sum_{k \in [K]} |s_{g,k}| \left(2 + \frac{8}{\varepsilon^2}\right) \Delta_{\max}$$

$$= |\cup S_g| \left(2 + \frac{8}{\varepsilon^2}\right) \Delta_{\max}$$

Above all, for the first term in (9), we have

$$\mathbb{E}\left[\sum_{t=1}^T \mathbb{1}\{\neg B(t), \neg C(t)\} \Delta_{S_t}\right]$$

$$\leq \sum_{s \neq s_{g,1}} \max_{k: s \notin S_{g,k}} \frac{6B^2 |s|^2 \Delta_s^{\max} \log T}{\left(\Delta_{s,k} - 2B |\cup S_g| \varepsilon\right)^2} + \left(\sum_{k \in [K]} \frac{C}{\varepsilon^2}\left(\frac{C'}{\varepsilon^4}\right)^{|s_{g,k}|} + |\cup S_g| \left(2 + \frac{8}{\varepsilon^2}\right)\right) \Delta_{\max}.$$

**The second term in** (9):

$$\mathbb{E}\left[\sum_{t=1}^T \mathbb{1}\{B(t)\} \Delta_{S_t}\right]$$

$$\leq \mathbb{E}\left[\sum_{t=1}^{T} \mathbb{1}\left\{\exists i \in [m] : |\theta_{t,i} - \hat{\mu}_{t,i}| > \sqrt{\frac{3\log T}{2N_{t,i}}}\right\}\right] \cdot \Delta_{\max}$$

$$\leq \sum_{i \in [m]} \mathbb{E}\left[\sum_{t=1}^{T} \mathbb{1}\left\{|\theta_{t,i} - \hat{\mu}_{t,i}| > \sqrt{\frac{3\log T}{2N_{t,i}}}\right\}\right] \cdot \Delta_{\max}$$

$$= \sum_{i \in [m]} \sum_{t=1}^{T} \sum_{w=1}^{T-1} \mathbb{P}\left(N_{t,i} = w, |\theta_{t,i} - \hat{\mu}_{t,i}| > \sqrt{\frac{3\log T}{2N_{t,i}}}\right) \cdot \Delta_{\max}$$

$$= \sum_{i \in [m]} \sum_{t=1}^{T} \sum_{w=1}^{T-1} \mathbb{P}\left(N_{t,i} = w\right) \cdot \mathbb{P}\left(|\theta_{t,i} - \hat{\mu}_{t,i}| > \sqrt{\frac{3\log T}{2N_{t,i}}} \mid N_{t,i} = w\right) \cdot \Delta_{\max}$$

$$\leq \sum_{i \in [m]} \sum_{t=1}^{T} \sum_{w=1}^{T-1} \mathbb{P}\left(N_{t,i} = w\right) \cdot 2\exp\left(-3\log T\right) \cdot \Delta_{\max} \tag{10}$$

$$\leq \sum_{i \in [m]} \sum_{t=1}^{T} \frac{2}{T} \cdot \Delta_{\max}$$

$$= 2m\Delta_{\max},$$

where (10) comes from the result of Lemma 7.

**The third term in** (9):

$$\mathbb{E}\left[\sum_{t=1}^{T} \mathbb{1}\{C(t)\}\Delta_{S_t}\right]$$

$$\leq \mathbb{E}\left[\sum_{t=1}^{T} \mathbb{1}\left\{\exists i \in [m] : |\hat{\mu}_{t,i} - \mu_i| > \sqrt{\frac{3\log T}{2N_{t,i}}}\right\}\right] \cdot \Delta_{\max}$$

$$\leq \sum_{i \in [m]} \mathbb{E}\left[\sum_{t=1}^{T} \mathbb{1}\left\{|\hat{\mu}_{t,i} - \mu_i| > \sqrt{\frac{3\log T}{2N_{t,i}}}\right\}\right] \cdot \Delta_{\max}$$

$$= \sum_{i \in [m]} \sum_{t=1}^{T} \sum_{w=1}^{T-1} \mathbb{P}\left(N_{t,i} = w, |\hat{\mu}_{t,i} - \mu_i| > \sqrt{\frac{3\log T}{2N_{t,i}}}\right) \cdot \Delta_{\max}$$

$$= \sum_{i \in [m]} \sum_{t=1}^{T} \sum_{w=1}^{T-1} \mathbb{P}\left(N_{t,i} = w\right) \cdot \mathbb{P}\left(|\hat{\mu}_{t,i} - \mu_i| > \sqrt{\frac{3\log T}{2N_{t,i}}} \mid N_{t,i} = w\right) \cdot \Delta_{\max}$$

$$\leq \sum_{i \in [m]} \sum_{t=1}^{T} \sum_{w=1}^{T-1} \mathbb{P}\left(N_{t,i} = w\right) \cdot 2\exp\left(-3\log T\right) \cdot \Delta_{\max} \tag{11}$$

$$\leq \sum_{i \in [m]} \sum_{t=1}^{T} \frac{2}{T} \cdot \Delta_{\max}$$

$$= 2m\Delta_{\max},$$

where (11) is obtained by Lemma 5.

Combine these three terms in (9), we can get the result in Theorem 2,

$$R_g(T) \leq \sum_{s \neq s_{g,1}} \max_{k:s \notin S_{g,k}} \frac{6B^2 |s|^2 \Delta_s^{\max} \log T}{\left(\Delta_{s,k} - 2B |\cup S_g|\varepsilon\right)^2} + \sum_{k \in [K]} \frac{C}{\varepsilon^2}\left(\frac{C'}{\varepsilon^4}\right)^{|s_{g,k}|} \Delta_{\max}$$

$$+ \left(|\cup S_g|\left(2 + \frac{8}{\varepsilon^2}\right) + 4m\right)\Delta_{\max}.$$

### C.1 Technical Lemmas

**Lemma 1.** *In Algorithm 2, for any $k \in [K]$, we have*

$$\mathbb{E}\left[\sum_{t=1}^{T}\mathbb{1}\left\{\neg B(t), \neg C(t), S_{t,k-1} = S_{g,k-1}, \left\|\theta_{t,S_{g,k-1}} - \mu_{S_{g,k-1}}\right\|_{\infty} \le \varepsilon, s_{t,k} \ne s_{g,k}\right\}\Delta_{S_t}\right]$$

$$\le \mathbb{E}\left[\sum_{s \notin S_{g,k}}\sum_{t=1}^{T}\mathbb{1}\{s = s_{t,k}, N_{t,s} \le L(s)\}\Delta_{S_t}\right] + \frac{C}{\varepsilon^2}\left(\frac{C'}{\varepsilon^4}\right)^{|s_{g,k}|}\Delta_{\max},$$

*where $C, C'$ are two universal constants.*

*Proof.* Recall for any unit $s \notin S_{g,k}$, $\Delta_{s,k} = r(S_{g,k}, \mu) - r(S_{g,k-1} \cup \{s\}, \mu)$. Define the event

$$D_k(t) = \left\{B\sum_{i \in s_{t,k}}|\theta_{t,i} - \mu_i| > \Delta_{s_{t,k},k} - B\left(2\sum_{k'<k}|s_{g,k'}| + |s_{g,k}| + 1\right)\varepsilon\right\}.$$

Then the formula in Lemma 1 can further bounded by

$$\mathbb{E}\left[\sum_{t=1}^{T}\mathbb{1}\left\{\neg B(t), \neg C(t), S_{t,k-1} = S_{g,k-1}, \left\|\theta_{t,S_{g,k-1}} - \mu_{S_{g,k-1}}\right\|_{\infty} \le \varepsilon, s_{t,k} \ne s_{g,k}\right\}\Delta_{S_t}\right]$$

$$\le \mathbb{E}\left[\sum_{t=1}^{T}\mathbb{1}\left\{\neg B(t), \neg C(t), S_{t,k-1} = S_{g,k-1}, \left\|\theta_{t,S_{g,k-1}} - \mu_{S_{g,k-1}}\right\|_{\infty} \le \varepsilon, s_{t,k} \ne s_{g,k}, D_k(t)\right\}\Delta_{S_t}\right]$$

$$(12)$$

$$+ \mathbb{E}\left[\sum_{t=1}^{T}\mathbb{1}\left\{S_{t,k-1} = S_{g,k-1}, \left\|\theta_{t,S_{g,k-1}} - \mu_{S_{g,k-1}}\right\|_{\infty} \le \varepsilon, s_{t,k} \ne s_{g,k}, \neg D_k(t)\right\}\right]\Delta_{\max}.$$

$$(13)$$

For term (12), we claim that the event $\left\{\neg B(t), \neg C(t), S_{t,k-1} = S_{g,k-1}, \left\|\theta_{t,S_{g,k-1}} - \mu_{S_{g,k-1}}\right\|_{\infty} \le \varepsilon, s_{t,k} \ne s_{g,k}, D_k(t)\right\}$ implies $N_{t,s_{t,k}} \le L(s_{t,k})$. This claim can be proved by contradiction.

Suppose $N_{t,s_{t,k}} > L(s_{t,k})$, then we must have

$$B\sum_{i \in s_{t,k}}|\theta_{t,i} - \mu_i| \le B\sum_{i \in s_{t,k}}\sqrt{\frac{6\log T}{N_{t,s_{t,k}}}}$$

$$< B|s_{t,k}|\sqrt{\frac{6\log T}{6B^2|s_{t,k}|^2\log T}}\left(\Delta_{s_{t,k},k} - 2B|\cup S_g|\varepsilon\right)$$

$$\le B|s_{t,k}|\sqrt{\frac{\log T}{B^2|s_{t,k}|^2\log T}}\left(\Delta_{s_{t,k},k} - B\left(2\sum_{k'<k}|s_{g,k'}| + |s_{g,k}| + 1\right)\varepsilon\right)$$

$$= \Delta_{s_{t,k},k} - B\left(2\sum_{k'<k}|s_{g,k'}| + |s_{g,k}| + 1\right)\varepsilon,$$

where the first inequality is due to the event of $\neg B(t)$ and $\neg C(t)$, the second inequality comes from the fact $N_{t,s_{t,k}} > L(s_{t,k})$ and the definition of $L(s_{t,k})$. Thus we conclude the event $D_k(t)$ will not happen and the claim is proved.

Then according to the above claim, there is

$$(12) \le \mathbb{E}\left[\sum_{t=1}^{T}\mathbb{1}\left\{N_{t,s_{t,k}} \le L(s_{t,k}), s_{t,k} \notin S_{g,k}\right\}\Delta_{S_t}\right]$$

$$\le \mathbb{E}\left[\sum_{s \notin S_{g,k}}\sum_{t=1}^{T}\mathbb{1}\{s = s_{t,k}, N_{t,s} \le L(s)\}\Delta_{S_t}\right].$$

For term (13), we first define event $\mathcal{E}_{k,1}(t)$ as

$$\mathcal{E}_{k,1}(t) = \Big\{ \forall \theta' \text{ with } \theta'_i = \theta_{t,i} \text{ for any } i \notin s_{g,k} \text{ and } \left\| \theta'_{s_{g,k}} - \mu_{s_{g,k}} \right\|_\infty \leq \varepsilon, \text{ then } s_{g,k} \text{ is the } k\text{-th}$$

$$\text{selected unit by } \texttt{Greedy} \text{ when the input is } \theta' \Big\}$$

and the event $\mathcal{E}_{k,2}(t)$ as

$$\mathcal{E}_{k,2}(t) = \left\{ \left\| \theta_{t,s_{g,k}} - \mu_{s_{g,k}} \right\|_\infty > \varepsilon \right\}.$$

We claim that if the event $\left\{ S_{t,k-1} = S_{g,k-1}, \left\| \theta_{t,S_{g,k-1}} - \mu_{S_{g,k-1}} \right\|_\infty \leq \varepsilon, s_{t,k} \neq s_{g,k}, \neg D_k(t) \right\}$ happens, then $\mathcal{E}_{k,1}(t)$ and $\mathcal{E}_{k,2}(t)$ hold.

We first consider event $\mathcal{E}_{k,1}(t)$. To show this event holds, it is sufficient to prove for any $\theta'$ satisfies the condition defined in $\mathcal{E}_{k,1}(t)$, $S_{g,k-1}$ is still the set of units selected by $\texttt{Greedy}$ in the first $k-1$ steps under $\theta'$ and for any $s' \notin S_{g,k}, r(S_{g,k-1} \cup \{s'\}, \theta') < r(S_{g,k}, \theta')$.

We now prove that for any $\theta'$ satisfying the condition defined in $\mathcal{E}_{k,1}(t)$, $S_{g,k-1}$ is still the set of units selected by $\texttt{Greedy}$ under $\theta'$ in the first $k-1$ steps. The event $S_{t,k-1} = S_{g,k-1}$ means that for any $k' < k, s \notin S_{g,k'}$, we have $r(S_{g,k'}, \theta_t) > r(S_{g,k'-1} \cup \{s\}, \theta_t)$. The mean vector $\theta'$ and $\theta_t$ are only different on $s_{g,k}$, thus for any $k' < k, s \notin S_{g,k'} \cup \{s_{g,k}\}$, we still have $r(S_{g,k'}, \theta') > r(S_{g,k'-1} \cup \{s\}, \theta')$. As for the unit $s_{g,k}$, for any $k' < k$, we have

$$\begin{aligned}
r(S_{g,k'}, \theta') =& r(S_{g,k'}, \theta_t) \\
\geq& r(S_{g,k'}, \mu) - B \left| \cup S_{g,k'} \right| \varepsilon \\
=& r(S_{g,k'-1} \cup \{s_{g,k}\}, \mu) + \Delta_{s_{g,k},k'} - B \left| \cup S_{g,k'} \right| \varepsilon \\
\geq& r(S_{g,k'-1} \cup \{s_{g,k}\}, \theta') - B \left( \left| \cup S_{g,k'-1} \right| + \left| s_{g,k} \right| \right) \varepsilon + \Delta_{s_{g,k},k'} - B \left| \cup S_{g,k'} \right| \varepsilon \\
\geq& r(S_{g,k'-1} \cup \{s_{g,k}\}, \theta') + \Delta_{s_{g,k},k'} - 2B \left| \cup S_g \right| \varepsilon \\
>& r(S_{g,k'-1} \cup \{s_{g,k}\}, \theta').
\end{aligned}$$

where the last inequality holds due to the requirement of $\varepsilon$ in Theorem 2. Above all, we conclude $\forall k' < k, s \notin S_{g,k'}, r(S_{g,k'}, \theta') > r(S_{g,k'-1} \cup \{s\}, \theta')$, thus $S_{g,k-1}$ is still the set of units selected by $\texttt{Greedy}$ in the first $k-1$ steps.

Next we prove for any $\theta'$ defined in $\mathcal{E}_{k,1}(t)$ and unit $s' \notin S_{g,k}, r(S_{g,k-1} \cup \{s'\}, \theta') < r(S_{g,k}, \theta')$.

$$\begin{aligned}
r(S_{g,k-1} \cup \{s'\}, \theta') =& r(S_{g,k-1} \cup \{s'\}, \theta_t) \\
\leq& r(S_{t,k}, \theta_t) \quad (\texttt{Greedy's property and } S_{t,k-1} = S_{g,k-1}) \\
\leq& r(S_{t,k}, \mu) + B \sum_{i \in \cup S_{t,k-1}} |\theta_{t,i} - \mu_i| + B \sum_{i \in s_{t,k}} |\theta_{t,i} - \mu_i| \quad (\text{Lipschitz continuity}) \\
\leq& r(S_{t,k}, \mu) + B \sum_{k'<k} |s_{g,k'}| \varepsilon + B \sum_{i \in s_{t,k}} |\theta_{t,i} - \mu_i| \\
\leq& r(S_{t,k}, \mu) + B \sum_{k'<k} |s_{g,k'}| \varepsilon + \Delta_{s_{t,k},k} - B \left( 2 \sum_{k'<k} |s_{g,k'}| + |s_{g,k}| + 1 \right) \varepsilon \\
& \tag{14} \\
\leq& r(S_{g,k}, \mu) + B \sum_{k'<k} |s_{g,k'}| \varepsilon - B \left( 2 \sum_{k'<k} |s_{g,k'}| + |s_{g,k}| + 1 \right) \varepsilon \tag{15} \\
=& r(S_{g,k}, \mu) - B \left( \left| \cup S_{g,k} \right| + 1 \right) \varepsilon \\
\leq& r(S_{g,k}, \theta') + B \left| \cup S_{g,k} \right| \varepsilon - B \left( \left| \cup S_{g,k} \right| + 1 \right) \varepsilon \quad (\text{Lipschitz continuity}) \\
<& r(S_{g,k}, \theta'),
\end{aligned}$$

where the first equality is because $\theta'_i = \theta_{t,i}$ for any $i \in s'$ and $i \in \cup S_{g,k-1}$, the third inequality is because $S_{t,k-1} = S_{g,k-1}$ and $\left\| \theta_{t,S_{g,k-1}} - \mu_{S_{g,k-1}} \right\|_\infty \leq \varepsilon$. (14) comes from the definition of $\neg D_k(t)$ and (15) is due to the definition of $\Delta_{s_{t,k},k}$ and the fact $S_{t,k-1} = S_{g,k-1}$.

Above all, we have proved if event $\left\{S_{t,k-1} = S_{g,k-1}, \left\|\theta_{t,S_{g,k-1}} - \mu_{S_{g,k-1}}\right\|_\infty \leq \varepsilon, s_{t,k} \neq s_{g,k}, \neg D_k(t)\right\}$ happens, then $\mathcal{E}_{k,1}(t)$ holds.

Next we consider $\mathcal{E}_{k,2}(t)$. By contradiction, when the event $\left\{S_{t,k-1} = S_{g,k-1}, \left\|\theta_{t,S_{g,k-1}} - \mu_{S_{g,k-1}}\right\|_\infty \leq \varepsilon, s_{t,k} \neq s_{g,k}, \neg D_k(t)\right\}$ happens, if $\neg\mathcal{E}_{k,2}(t) = \left\{\left\|\theta_{t,s_{g,k}} - \mu_{s_{g,k}}\right\|_\infty \leq \varepsilon\right\}$ holds, then $\theta_t$ satisfies the property of $\theta'$ defined in $\mathcal{E}_{k,1}(t)$. Thus according to $\mathcal{E}_{k,1}(t)$, $s_{g,k}$ would be the $k$-th selected unit by Greedy when the input is $\theta_t$, or in other words $s_{g,k} = s_{t,k}$. This contradicts $\left\{S_{t,k-1} = S_{g,k-1}, \left\|\theta_{S_{g,k-1}}(t) - \mu_{S_{g,k-1}}\right\|_\infty \leq \varepsilon, s_{t,k} \neq s_{g,k}, \neg D_k(t)\right\}$. Thus we conclude $\mathcal{E}_{k,2}(t)$ also holds.

Above all, for term (13) we have

$$
\begin{aligned}
(13) &\leq \mathbb{E}\left[\sum_{t=1}^{T} \mathbb{1}\{\mathcal{E}_{k,1}(t), \mathcal{E}_{k,2}(t)\}\right]\Delta_{\max} \\
&\leq \sum_{q\geq 0}\mathbb{E}\left[\sum_{t=\tau_{k,q}+1}^{\tau_{k,q+1}} \mathbb{1}\{\mathcal{E}_{k,1}(t), \mathcal{E}_{k,2}(t)\}\right]\Delta_{\max} \\
&\leq \left(\sum_{q\geq 0}\mathbb{E}\left[\sup_{t\geq\tau_{k,q}+1}\prod_{i\in s_{g,k}}\frac{1}{\mathbb{P}\left(|\theta_{t,i}-\mu_i|\leq\varepsilon\mid\mathcal{H}_t\right)}\right] - 1\right)\Delta_{\max} \\
&\leq \left(\sum_{q=0}^{\lceil 8/\varepsilon^2\rceil - 1}\left(c\varepsilon^{-4}\right)^{|s_{g,k}|} + \sum_{q\geq\lceil 8/\varepsilon^2\rceil} e^{-\varepsilon^2 q/8}\left(c'\varepsilon^{-4}\right)^{|s_{g,k}|}\right)\Delta_{\max} \\
&\leq \frac{C}{\varepsilon^2}\left(\frac{C'}{\varepsilon^4}\right)^{|s_{g,k}|}\Delta_{\max},
\end{aligned}
$$

where $\tau_{k,q}$ is the round at which $\mathcal{E}_{k,1}(t) \wedge \neg\mathcal{E}_{k,2}(t)$ occurs for the $q$-th time, note $\tau_{k,0} = 0$ for any $k \in [K]$. The third inequality comes from the result in Lemma 2 and the fourth comes from the Lemma 8. Here $C, C'$ are two universal constants. $\qquad\square$

**Lemma 2.** *Let $\tau_{k,q}$ be the round at which $\mathcal{E}_{k,1}(t) \wedge \neg\mathcal{E}_{k,2}(t)$ occurs for the $q$-th time, let $\tau_{k,0} = 0$ for any $k \in [K]$. Then for Algorithm 2, we have*

$$
\mathbb{E}\left[\sum_{t=\tau_{k,q}+1}^{\tau_{k,q+1}} \mathbb{1}\{\mathcal{E}_{k,1}(t), \mathcal{E}_{k,2}(t)\}\right] \leq \mathbb{E}\left[\sup_{t\geq\tau_{k,q}+1}\prod_{i\in s_{g,k}}\frac{1}{\mathbb{P}\left(|\theta_{t,i}-\mu_i|\leq\varepsilon\mid\mathcal{H}_t\right)}\right] - 1.
$$

*Proof.* Conditioned on history $\mathcal{H}_t$, the event $\mathcal{E}_{k,1}(t)$ and $\mathcal{E}_{k,2}(t)$ are independent. Thus

$$
\mathbb{E}\left[\sum_{t=\tau_{k,q}+1}^{\tau_{k,q+1}} \mathbb{1}\{\mathcal{E}_{k,1}(t), \mathcal{E}_{k,2}(t)\}\right] = \mathbb{E}\left[\sum_{q'\geq 1}(q'-1)\mathbb{P}\left(\neg\mathcal{E}_{k,2}(\tau_{k,q,q'})\mid\mathcal{H}_{\tau_{k,q,q'}}\right)\prod_{j=1}^{q'-1}\mathbb{P}\left(\mathcal{E}_{k,2}(\tau_{k,q,j})\mid\mathcal{H}_{\tau_{k,q,j}}\right)\right],
$$

here $\tau_{k,q,j}$ is the round when the event $\mathcal{E}_{k,1}(t)$ has happened for the $j$-th time after round $\tau_{k,q} + 1$.

The right hand side of the above equality is the expectation of a time-varying geometric distribution with the success probability of the $j$-th trial being $\mathbb{P}\left(\neg\mathcal{E}_{k,2}(\tau_{k,q,j})\mid\mathcal{H}_{\tau_{k,q,j}}\right)$. We can lower bound this probability by

$$
\inf_{t\geq\tau_{k,q}+1}\mathbb{P}\left(\neg\mathcal{E}_{k,2}(t)\mid\mathcal{H}_t\right) = \inf_{t\geq\tau_{k,q}+1}\prod_{i\in s_{g,k}}\mathbb{P}\left(|\theta_{t,i}-\mu_i|\leq\varepsilon\mid\mathcal{H}_t\right).
$$

Then according to the monotonicity of the expectation, we can upper bound the above expectation by

$$
\mathbb{E}\left[\sum_{q'\geq 1}(q'-1)\mathbb{P}\left(\neg\mathcal{E}_{k,2}(\tau_{k,q,q'})\mid\mathcal{H}_{\tau_{k,q,q'}}\right)\prod_{j=1}^{q'-1}\mathbb{P}\left(\mathcal{E}_{k,2}(\tau_{k,q,j})\mid\mathcal{H}_{\tau_{k,q,j}}\right)\right]
$$

$$\leq \mathbb{E}\left[\sup_{t \geq \tau_{k,q}+1} \frac{1}{\prod_{i \in s_{g,k}} \mathbb{P}\left(|\theta_{t,i} - \mu_i| \leq \varepsilon \mid \mathcal{H}_t\right)}\right] - 1.$$

$\square$

**Lemma 3.** *In Algorithm 2, for any $k \in [K]$, we have*

$$\mathbb{E}\left[\sum_{t=1}^T \mathbb{1}\left\{s_{t,k} = s_{g,k}, \left\|\theta_{t,s_{g,k}} - \mu_{s_{g,k}}\right\|_\infty > \varepsilon\right\}\right] \leq |s_{g,k}|\left(2 + \frac{8}{\varepsilon^2}\right).$$

*Proof.* We first decompose the event as

$$\mathbb{E}\left[\sum_{t=1}^T \mathbb{1}\left\{s_{t,k} = s_{g,k}, \left\|\theta_{t,s_{g,k}} - \mu_{s_{g,k}}\right\|_\infty > \varepsilon\right\}\right]$$

$$\leq \mathbb{E}\left[\sum_{t=1}^T \mathbb{1}\left\{s_{t,k} = s_{g,k}, \left\|\hat{\mu}_{t,s_{g,k}} - \mu_{s_{g,k}}\right\|_\infty > \frac{\varepsilon}{2}\right\}\right] \tag{16}$$

$$+ \mathbb{E}\left[\sum_{t=1}^T \mathbb{1}\left\{s_{t,k} = s_{g,k}, \left\|\theta_{t,s_{g,k}} - \hat{\mu}_{t,s_{g,k}}\right\|_\infty > \frac{\varepsilon}{2}\right\}\right]. \tag{17}$$

For term (16), we have

$$(16) = \mathbb{E}\left[\sum_{t=1}^T \mathbb{1}\left\{s_{t,k} = s_{g,k}, \exists i \in s_{g,k} : |\hat{\mu}_{t,i} - \mu_i| > \frac{\varepsilon}{2}\right\}\right]$$

$$\leq \sum_{i \in s_{g,k}} \mathbb{E}\left[\sum_{t=1}^T \mathbb{1}\left\{i \in \cup S_t : |\hat{\mu}_{t,i} - \mu_i| > \frac{\varepsilon}{2}\right\}\right]$$

$$= \sum_{i \in s_{g,k}} \mathbb{E}\left[\sum_{w=0}^T \mathbb{E}\left[\sum_{t \in [T] : N_{t,i} = w} \mathbb{1}\left\{i \in \cup S_t : |\hat{\mu}_{t,i} - \mu_i| > \frac{\varepsilon}{2}\right\}\right]\right]$$

$$\leq \sum_{i \in s_{g,k}} \mathbb{E}\left[\sum_{w=0}^T \mathbb{P}\left(|\hat{\mu}_{t,i} - \mu_i| > \frac{\varepsilon}{2} \text{ for } t \text{ satisfies } N_{t,i} = w \text{ and } N_{t+1,i} = w + 1\right)\right]$$

$$\leq \sum_{i \in s_{g,k}} \left(1 + \sum_{w=1}^T \mathbb{P}\left(|\hat{\mu}_{t,i} - \mu_i| > \frac{\varepsilon}{2} \text{ for } t \text{ satisfies } N_{t,i} = w \text{ and } N_{t+1,i} = w + 1\right)\right)$$

$$\leq \sum_{i \in s_{g,k}} \left(1 + 2\sum_{w=1}^T \exp\left(-\frac{w\varepsilon^2}{2}\right)\right) \tag{18}$$

$$\leq |s_{g,k}|\left(1 + 2\sum_{w=1}^\infty \left(\exp\left(-\frac{\varepsilon^2}{2}\right)\right)^w\right)$$

$$\leq |s_{g,k}|\left(1 + 2\frac{\exp\left(-\frac{\varepsilon^2}{2}\right)}{1 - \exp\left(-\frac{\varepsilon^2}{2}\right)}\right)$$

$$\leq |s_{g,k}|\left(1 + \frac{4}{\varepsilon^2}\right),$$

where (18) is due to the Lemma 5. Similarly, we have the following bound for term (17),

$$(17) = \mathbb{E}\left[\sum_{t=1}^T \mathbb{1}\left\{s_{t,k} = s_{g,k}, \exists i \in s_{g,k} : |\hat{\mu}_{t,i} - \theta_{t,i}| > \frac{\varepsilon}{2}\right\}\right]$$

$$\leq \sum_{i \in s_{g,k}} \mathbb{E}\left[\sum_{t=1}^{T} \mathbb{1}\left\{i \in \cup S_t : |\hat{\mu}_{t,i} - \theta_{t,i}| > \frac{\varepsilon}{2}\right\}\right]$$

$$= \sum_{i \in s_{g,k}} \mathbb{E}\left[\sum_{w=0}^{T} \mathbb{E}\left[\sum_{t \in [T]: N_{t,i} = w} \mathbb{1}\left\{i \in \cup S_t : |\hat{\mu}_{t,i} - \theta_{t,i}| > \frac{\varepsilon}{2}\right\}\right]\right]$$

$$\leq \sum_{i \in s_{g,k}} \mathbb{E}\left[\sum_{w=0}^{T} \mathbb{P}\left(|\hat{\mu}_{t,i} - \theta_{t,i}| > \frac{\varepsilon}{2} \text{ for } t \text{ satisfies } N_{t,i} = w \text{ and } N_{t+1,i} = w + 1\right)\right]$$

$$\leq \sum_{i \in s_{g,k}} \left(1 + \sum_{w=1}^{T} \mathbb{P}\left(|\hat{\mu}_{t,i} - \theta_{t,i}| > \frac{\varepsilon}{2} \text{ for } t \text{ satisfies } N_{t,i} = w \text{ and } N_{t+1,i} = w + 1\right)\right)$$

$$\leq \sum_{i \in s_{g,k}} \left(1 + 2 \sum_{w=1}^{T} \exp\left(-\frac{w\varepsilon^2}{2}\right)\right) \tag{19}$$

$$\leq |s_{g,k}| \left(1 + 2 \sum_{w=1}^{\infty} \left(\exp\left(-\frac{\varepsilon^2}{2}\right)\right)^{w}\right)$$

$$\leq |s_{g,k}| \left(1 + 2 \frac{\exp\left(-\frac{\varepsilon^2}{2}\right)}{1 - \exp\left(-\frac{\varepsilon^2}{2}\right)}\right)$$

$$\leq |s_{g,k}| \left(1 + \frac{4}{\varepsilon^2}\right),$$

where (19) is due to the Lemma 7.

Above all, we have

$$(16) + (17) \leq |s_{g,k}| \left(1 + \frac{4}{\varepsilon^2}\right) + |s_{g,k}| \left(1 + \frac{4}{\varepsilon^2}\right)$$

$$= |s_{g,k}| \left(2 + \frac{8}{\varepsilon^2}\right).$$

$\square$

# D   Analysis of CTS with Gaussian Priors and `Greedy` Oracle

**Theorem 3.** *When the reward distribution for each arm $i$ is $D_i = \mathcal{N}(\mu_i, 1)$, the cumulative greedy regret of Algorithm 3 can be upper bounded by*

$$R_g(T) \leq \sum_{s \neq s_{g,1}} \max_{k: s \notin S_{g,k}} \frac{8B^2 |s|^2 \Delta_s^{\max} \log T}{(\Delta_{s,k} - 2B |\cup S_g| \varepsilon)^2} + \sum_{k \in [K]} \frac{C}{\varepsilon^2} \left(\frac{C'}{\varepsilon^4}\right)^{|s_{g,k}|} \Delta_{\max}$$

$$+ \left(|\cup S_g| \left(2 + \frac{8}{\varepsilon^2}\right) + m\right) \Delta_{\max}$$

$$= O\left(\sum_{s \neq s_{g,1}} \max_{k: s \notin S_{g,k}} \frac{B^2 |s|^2 \Delta_{\max} \log T}{\Delta_{s,k}^2}\right),$$

*for any $\varepsilon$ such that $\forall s \neq s_{g,1}$ and $k$ satisfying $s \notin S_{g,k}$, $\Delta_{s,k} > 2B |\cup S_g| \varepsilon$, where $B$ is the coefficient of the Lipschitz continuity condition, $|\cup S_g|$ is the number of base arms that belong to the units contained in $S_g$, $C$ and $C'$ are two universal constants.*

*Proof.* The proof of Theorem 3 is very similar to that of Theorem 2 in Appendix C, here we only state the differences. When proving Theorem 3, we redefine $L(s)$, $B(t)$ and $C(t)$ with slight differences

in constants as

$$L(s) = \max_{k:s\notin S_{g,k}} \frac{8B^2\,|s|^2\log T}{\left(\Delta_{s,k}-2B\,|\cup S_g|\,\varepsilon\right)^2}\,,$$

$$B(t) = \left\{\exists i\in[m]:|\theta_{t,i}-\hat{\mu}_{t,i}|>\sqrt{\frac{2\log T}{N_{t,i}}}\right\}\,,$$

$$C(t) = \left\{\exists i\in[m]:|\hat{\mu}_{t,i}-\mu_i|>\sqrt{\frac{2\log T}{N_{t,i}}}\right\}\,.$$

Based on these new definitions, we use the result of Lemma 6 instead of the result of Lemma 5 and 7 to get an upper bound for Eq (10) and (11). Besides, when proving Lemma 1, we use the upper bound for [24, Eq(4)] instead of using the result of Lemma 8. □

## E  Regret Analysis for the Case of Multiple `Greedy` Solutions under $\mu$

In this section, we discuss how to extend the proof of Theorem 2 to the case where multiple solutions can be returned by `Greedy` with input $\mu$, or equivalently the optimal unit in each step $k$ (Line 4 in Algorithm 1) may be not unique.

For any $k\in[K]$, define

$$\sigma_k = \left\{\{s_1,s_2,\ldots,s_k\}: s_1\in\operatorname*{argmax}_s r(\{s\},\mu),\ldots,s_k\in\operatorname*{argmax}_{s\notin\{s_1,\ldots,s_{k-1}\}} r(\{s_1,\ldots,s_{k-1},s\},\mu)\right\}$$

as the set of actions containing $k$ units that could be selected by `Greedy` in the first $k$ steps. And denote

$$\bar{\sigma}_k = \{s_k:\{s_1,s_2,\ldots,s_k\}\in\sigma_k\}$$

as the set of units which may be selected as the $k$-th unit by `Greedy` under $\mu$. Let $\sigma_0=\emptyset$. Then for any $k$ and $S\in\sigma_{k-1}$, define

$$\sigma_k(S) = \left\{s: s\in\operatorname*{argmax}_{s\notin S} r(S\cup\{s\},\mu)\right\}$$

as the set of units that may be selected by `Greedy` in the $k$-th step when $S$ is selected in the previous $k-1$ steps. Let $\sigma_1(\emptyset)=\{s:s\in\operatorname*{argmax}_{s\in\mathcal{U}} r(\{s\},\mu)\}$. Denote $s_{g,S,k}\in\operatorname*{argmin}_{s\in\sigma_k(S)}|s|$ as one of the optimal $k$-th unit when $S$ is selected with the minimum unit size. And for any $s\notin S\cup\sigma_k(S)$, define $\Delta_{s,S,k}=r(S\cup\{s_{g,S,k}\},\mu)-r(S\cup\{s\},\mu)>0$ as the corresponding reward gap. Let $S'_g\in\operatorname*{argmax}_{S\in\sigma_K}|\cup S|$ be one of the possible actions returned by `Greedy` containing the maximum number of base arms. Then for any unit $s$, define

$$L(s) = \max_{k\in[K],S\in\sigma_{k-1}:s\notin\sigma_k(S)} \frac{6B^2|s|^2\log T}{\left(\Delta_{s,S,k}-2B\,\left|\cup S'_g\right|\varepsilon\right)^2}\,.$$

Note $L(s)=0$ if $\{k\in[K],S\in\sigma_{k-1}:s\notin\sigma_k(S)\}=\emptyset$.

In this case, we regard how `Greedy` breaks the tie at each step as a black box. In order to take into account the worst case where `Greedy` always return a solution with minimum reward compared to other possible solutions under $\mu$, we define $S_g\in\operatorname*{argmin}_{S\in\sigma_K} r(S,\mu)$ as one of the possible actions returned by `Greedy` with the minimum expected reward, and define the cumulative greedy regret as

$$R_g(T) = \mathbb{E}\left[\sum_{t=1}^{T}\Delta_{S_t}\right] = \mathbb{E}\left[\sum_{t=1}^{T}\max\{r(S_g,\mu)-r(S_t,\mu),0\}\right]\,.$$

**Theorem 4.** *When there are multiple* `Greedy` *solutions with input* $\mu$*, the cumulative greedy regret of Algorithm 2 can be bounded by*

$$R_g(T) \le \sum_s \max_{k\in[K],S\in\sigma_{k-1}:s\notin\sigma_k(S)} \frac{6B^2|s|^2\Delta_s^{\max}\log T}{\left(\Delta_{s,S,k}-2B\,\left|\cup S'_g\right|\varepsilon\right)^2}$$

$$+ \left( 4m + \sum_{k \in [K]} \sum_{s \in \bar{\sigma}_k} |s| \left( 2 + \frac{8}{\varepsilon^2} \right) + \sum_{k \in [K]} \sum_{S \in \sigma_{k-1}} \frac{C}{\varepsilon^2} \left( \frac{C'}{\varepsilon^4} \right)^{|s_{g,S,k}|} \right) \Delta_{\max}.$$

*for any $\varepsilon$ such that $\forall k \in [K], S \in \sigma_{k-1}, s \notin \sigma_k(S)$, there is $\Delta_{s,S,k} > 2B \left| \cup S'_g \right| \varepsilon$, where $B$ is the coefficient of the Lipschitz continuity and $|\cup S|$ is the number of base arms that belong to the units contained in $S$, $C$ and $C'$ are two universal constants.*

*Proof.* With the same definition of $B(t)$ and $C(t)$ as in Section C, the greedy regret can be decomposed by

$$R_g(T) \leq \mathbb{E} \left[ \sum_{t=1}^{T} \mathbb{1}\{B(t)\} \Delta_{S_t} \right] + \mathbb{E} \left[ \sum_{t=1}^{T} \mathbb{1}\{C(t)\} \Delta_{S_t} \right] \mathbb{E} \left[ \sum_{t=1}^{T} \mathbb{1}\{\neg B(t), \neg C(t)\} \Delta_{S_t} \right]. \quad (20)$$

Same as the proof for the second and the third term in (9), we have the following bound for the first and the second term in (20),

$$\mathbb{E} \left[ \sum_{t=1}^{T} \mathbb{1}\{B(t)\} \Delta_{S_t} \right] + \mathbb{E} \left[ \sum_{t=1}^{T} \mathbb{1}\{C(t)\} \Delta_{S_t} \right] \leq 4m\Delta_{\max}.$$

The main difference is to bound the last term. Intuitively, if $S_t \in \sigma_K$, we would have $\Delta_{S_t} = 0$. Thus we only need to consider the case where $S_t \notin \sigma_K$ to bound the regret. We will analyze such case by sequencially analyzing the $s_{t,k}$ for each $k = 1, 2, \ldots, K$. According to this idea, the first term can be bounded by

$$\mathbb{E} \left[ \sum_{t=1}^{T} \mathbb{1}\{\neg B(t), \neg C(t)\} \Delta_{S_t} \right]$$

$$\leq \sum_{k \in [K]} \mathbb{E} \left[ \sum_{t=1}^{T} \mathbb{1}\left\{\neg B(t), \neg C(t), S_{t,k-1} \in \sigma_{k-1}, \left\| \theta_{t,S_{t,k-1}} - \mu_{S_{t,k-1}} \right\|_\infty \leq \varepsilon, s_{t,k} \notin \sigma_k(S_{t,k-1}) \right\} \Delta_{S_t} \right]$$

$$+ \sum_{k \in [K]} \mathbb{E} \left[ \sum_{t=1}^{T} \mathbb{1}\left\{ s_{t,k} \in \bar{\sigma}_k, \left\| \theta_{t,s_{t,k}} - \mu_{s_{t,k}} \right\|_\infty > \varepsilon \right\} \right] \Delta_{\max}.$$

For the first term, According to Lemma 4, we have

$$\sum_{k \in [K]} \mathbb{E} \left[ \sum_{t=1}^{T} \mathbb{1}\left\{\neg B(t), \neg C(t), S_{t,k-1} \in \sigma_{k-1}, \left\| \theta_{t,S_{t,k-1}} - \mu_{S_{t,k-1}} \right\|_\infty \leq \varepsilon, s_{t,k} \notin \sigma_k(S_{t,k-1}) \right\} \Delta_{S_t} \right]$$

$$\leq \sum_{k \in [K]} \mathbb{E} \left[ \sum_s \sum_{t=1}^{T} \mathbb{1}\{s = s_{t,k}, N_{t,s} \leq L(s)\} \Delta_{S_t} \right] + \sum_{k \in [K]} \sum_{S \in \sigma_{k-1}} \frac{C}{\varepsilon^2} \left( \frac{C'}{\varepsilon^4} \right)^{|s_{g,S,k}|} \Delta_{\max}$$

$$\leq \mathbb{E} \left[ \sum_s \sum_{t=1}^{T} \mathbb{1}\{s \in S_t, N_{t,s} \leq L(s)\} \Delta_{S_t} \right] + \sum_{k \in [K]} \sum_{S \in \sigma_{k-1}} \frac{C}{\varepsilon^2} \left( \frac{C'}{\varepsilon^4} \right)^{|s_{g,S,k}|} \Delta_{\max}$$

$$\leq \sum_s L(s) \Delta_s^{\max} + \sum_{k \in [K]} \sum_{S \in \sigma_{k-1}} \frac{C}{\varepsilon^2} \left( \frac{C'}{\varepsilon^4} \right)^{|s_{g,S,k}|} \Delta_{\max}$$

$$\leq \sum_s \max_{k \in [K], S \in \sigma_{k-1}: s \notin \sigma_k(S)} \frac{6B^2|s|^2 \Delta_s^{\max} \log T}{\left( \Delta_{s,S,k} - 2B \left| \cup S'_g \right| \varepsilon \right)^2} + \sum_{k \in [K]} \sum_{S \in \sigma_{k-1}} \frac{C}{\varepsilon^2} \left( \frac{C'}{\varepsilon^4} \right)^{|s_{g,S,k}|} \Delta_{\max},$$

where $C, C'$ are two universal constants.

For the second term, we have

$$\sum_{k\in[K]} \mathbb{E}\left[\sum_{t=1}^{T} \mathbb{1}\left\{s_{t,k} \in \bar{\sigma}_k, \left\|\theta_{t,s_{t,k}} - \mu_{s_{t,k}}\right\|_\infty > \varepsilon\right\}\right] \leq \sum_{k\in[K]}\sum_{s\in\bar{\sigma}_k} \mathbb{E}\left[\mathbb{1}\left\{s_{t,k} = s, \left\|\theta_{t,s} - \mu_s\right\|_\infty > \varepsilon\right\}\right]$$

$$\leq \sum_{k\in[K]}\sum_{s\in\bar{\sigma}_k} |s|\left(2 + \frac{8}{\varepsilon^2}\right),$$

where the last inequality is obtained by applying the result of Lemma 3.

Above all, we have the following upper bound for the greedy regret

$$R_g(T) \leq \sum_s \max_{k\in[K], S\in\sigma_{k-1}:s\notin\sigma_k(S)} \frac{6B^2|s|^2\Delta_s^{\max}\log T}{\left(\Delta_{s,S,k} - 2B\left|\cup S_g'\right|\varepsilon\right)^2}$$

$$+ \left(4m + \sum_{k\in[K]}\sum_{s\in\bar{\sigma}_k} |s|\left(2 + \frac{8}{\varepsilon^2}\right) + \sum_{k\in[K]}\sum_{S\in\sigma_{k-1}} \frac{C}{\varepsilon^2}\left(\frac{C'}{\varepsilon^4}\right)^{|s_{g,S,k}|}\right)\Delta_{\max}.$$

$\square$

**Lemma 4.** *For Algorithm 2, when there are multiple* Greedy *solutions with input $\mu$, for any $k \in [K]$, we have*

$$\mathbb{E}\left[\sum_{t=1}^{T} \mathbb{1}\left\{\neg B(t), \neg C(t), S_{t,k-1}\in\sigma_{k-1}, \left\|\theta_{t,S_{t,k-1}} - \mu_{S_{t,k-1}}\right\|_\infty \leq \varepsilon, s_{t,k}\notin\sigma_k(S_{t,k-1})\right\}\Delta_{S_t}\right]$$

$$\leq \mathbb{E}\left[\sum_s\sum_{t=1}^{T} \mathbb{1}\left\{s = s_{t,k}, N_{t,s} \leq L(s)\right\}\Delta_{S_t}\right] + \sum_{S\in\sigma_{k-1}} \frac{C}{\varepsilon^2}\left(\frac{C'}{\varepsilon^4}\right)^{|s_{g,S,k}|}\Delta_{\max},$$

*where $C, C'$ are two universal constants.*

*Proof.* Recall given $S \in \sigma_{k-1}$, for any unit $s \notin S\cup\sigma_k(S)$, $\Delta_{s,S,k} = r(S\cup\{s_{g,S,k}\},\mu) - r(S\cup\{s\},\mu)$. Define the event

$$D_k(t) = \left\{B\sum_{i\in s_{t,k}} |\theta_{t,i} - \mu_i| > \Delta_{s_{t,k},S_{t,k-1},k} - B\left(2\left|\cup S_{t,k-1}\right| + \left|s_{g,S_{t,k-1},k}\right| + 1\right)\varepsilon\right\}.$$

Then the formula in Lemma 1 can further bounded by

$$\mathbb{E}\left[\sum_{t=1}^{T} \mathbb{1}\left\{\neg B(t), \neg C(t), S_{t,k-1}\in\sigma_{k-1}, \left\|\theta_{t,S_{t,k-1}} - \mu_{S_{t,k-1}}\right\|_\infty \leq \varepsilon, s_{t,k}\notin\sigma_k(S_{t,k-1})\right\}\right]$$

$$\leq \mathbb{E}\left[\sum_{t=1}^{T} \mathbb{1}\left\{\neg B(t), \neg C(t), S_{t,k-1}\in\sigma_{k-1}, \left\|\theta_{t,S_{t,k-1}} - \mu_{S_{t,k-1}}\right\|_\infty \leq \varepsilon, s_{t,k}\notin\sigma_k(S_{t,k-1}), D_k(t)\right\}\Delta_{S_t}\right]$$

$$(21)$$

$$+ \mathbb{E}\left[\sum_{t=1}^{T} \mathbb{1}\left\{\neg B(t), \neg C(t), S_{t,k-1}\in\sigma_{k-1}, \left\|\theta_{t,S_{t,k-1}} - \mu_{S_{t,k-1}}\right\|_\infty \leq \varepsilon, s_{t,k}\notin\sigma_k(S_{t,k-1}), \neg D_k(t)\right\}\Delta_{\max}\right].$$

$$(22)$$

For term (21), we claim that the event

$$\left\{\neg B(t), \neg C(t), S_{t,k-1}\in\sigma_{k-1}, \left\|\theta_{t,S_{t,k-1}} - \mu_{S_{t,k-1}}\right\|_\infty \leq \varepsilon, s_{t,k}\notin\sigma_k(S_{t,k-1}), D_k(t)\right\}$$

implies $N_{t,s_{t,k}} \leq L(s_{t,k})$. This claim can be proved by contradiction.

Suppose $N_{t,s_{t,k}} > L(s_{t,k})$, then we must have

$$B\sum_{i\in s_{t,k}} |\theta_{t,i} - \mu_i| \leq B\sum_{i\in s_{t,k}} \sqrt{\frac{6\log T}{N_{t,s_{t,k}}}}$$

$$< B\left|s_{t,k}\right|\sqrt{\frac{6\log T}{6B^2\left|s_{t,k}\right|^2\log T}}\left(\Delta_{s_{t,k},S_{t,k-1},k}-2B\left|\cup S_g'\right|\varepsilon\right)$$

$$\leq B\left|s_{t,k}\right|\sqrt{\frac{\log T}{B^2\left|s_{t,k}\right|^2\log T}}\left(\Delta_{s_{t,k},S_{t,k-1},k}-B\left(2\left|\cup S_{t,k-1}\right|+\left|s_{g,S_{t,k-1},k}\right|+1\right)\varepsilon\right)$$

$$= \Delta_{s_{t,k},S_{t,k-1},k}-B\left(2\left|\cup S_{t,k-1}\right|+\left|s_{g,S_{t,k-1},k}\right|+1\right)\varepsilon\,,$$

where the first inequality is due to the event of $\neg B(t)$ and $\neg C(t)$, the second inequality is obtained by substituting $N_{t,s_{t,k}}$ with $L(s_{t,k})$ and the third one comes from the definition of $S_g'$ and the fact that $S_{t,k-1}\in\sigma_{k-1}$. Thus we conclude the event $D_k(t)$ will not happen and the claim is proved.

Then according to the above claim, there is

$$(21)\leq\mathbb{E}\left[\sum_{t=1}^{T}\mathbb{1}\left\{N_{t,s_{t,k}}\leq L(s_{t,k}),S_{t,k-1}\in\sigma_{k-1},s_{t,k}\notin\sigma_k(S_{t,k-1})\right\}\Delta_{S_t}\right]$$

$$\leq\mathbb{E}\left[\sum_{s}\sum_{t=1}^{T}\mathbb{1}\left\{s=s_{t,k},N_{t,s}\leq L(s)\right\}\Delta_{S_t}\right]\,.$$

For term (22), we first define event $\mathcal{E}_{S,k,1}(t)$ for $S\in\sigma_{k-1}$ as

$$\mathcal{E}_{S,k,1}(t)=\Big\{\forall\theta'\text{ with }\theta_i'=\theta_{t,i}\text{ for any }i\notin s_{g,S,k}\text{ and }\left\|\theta_{s_{g,S,k}}'-\mu_{s_{g,S,k}}\right\|_{\infty}\leq\varepsilon,\text{ then }s_{g,S,k}\text{ is the }k\text{-th}$$
$$\text{selected unit by }\texttt{Greedy}\text{ when the input is }\theta'\Big\}$$

and the event $\mathcal{E}_{S,k,2}(t)$ as

$$\mathcal{E}_{S,k,2}(t)=\left\{\left\|\theta_{t,s_{g,S,k}}-\mu_{s_{g,S,k}}\right\|_{\infty}>\varepsilon\right\}\,.$$

We claim that if the event $\left\{S_{t,k-1}\in\sigma_{k-1},\left\|\theta_{t,S_{t,k-1}}-\mu_{S_{t,k-1}}\right\|_{\infty}\leq\varepsilon,s_{t,k}\notin\sigma_k(S_{t,k-1}),\neg D_k(t)\right\}$ happens, then $\mathcal{E}_{S_{t,k-1},k,1}(t)$ and $\mathcal{E}_{S_{t,k-1},k,2}(t)$ hold.

To prove $\mathcal{E}_{S_{t,k-1},k,1}(t)$ holds, it is sufficient to prove for any $\theta'$ defined in $\mathcal{E}_{S_{t,k-1},k,1}(t)$, $S_{t,k-1}$ is still the set of units selected by $\texttt{Greedy}$ in the first $k-1$ steps with input $\theta'$ and for any unit $s'\notin S_{t,k-1}\cup\left\{s_{g,S_{t,k-1}}\right\}$, $r(S_{t,k-1}\cup\left\{s'\right\},\theta')<r\left(S_{t,k-1}\cup\left\{s_{g,S_{t,k-1},k}\right\},\theta'\right)$ holds.

We now prove that for any $\theta'$ satisfying the condition defined in $\mathcal{E}_{S_{t,k-1},k,1}(t)$, $S_{t,k-1}$ is still the set of units selected by $\texttt{Greedy}$ in the first $k-1$ steps with input $\theta'$. The event $S_{t,k-1}$ is selected in the first $k-1$ steps with input $\theta_t$ means that for any $k'<k,s\notin S_{t,k'}$, we have $r(S_{t,k'},\theta_t)>r(S_{t,k'-1}\cup\left\{s\right\},\theta_t)$. The mean vector $\theta'$ and $\theta_t$ are only different on $s_{g,S_{t,k-1},k}$, thus for any $k'<k,s\notin S_{t,k'}\cup\left\{s_{g,S_{t,k-1},k}\right\}$, we still have $r(S_{t,k'},\theta')>r(S_{t,k'-1}\cup\left\{s\right\},\theta')$. And for the unit $s_{g,S_{t,k-1},k}$, for any $k'<k$, we have

$$r(S_{t,k'},\theta')=r(S_{t,k'},\theta_t)$$
$$\geq r(S_{t,k'},\mu)-B\left|\cup S_{t,k'}\right|\varepsilon$$
$$= r(S_{t,k'-1}\cup\left\{s_{g,S_{t,k-1},k}\right\},\mu)+\Delta_{s_{g,S_{t,k-1},k},S_{t,k'-1},k'}-B\left|\cup S_{t,k'}\right|\varepsilon$$
$$\geq r(S_{t,k'-1}\cup\left\{s_{g,S_{t,k-1},k}\right\},\theta')-B\left(\left|\cup S_{t,k'}\right|+\left|\cup S_{t,k'-1}\right|+\left|s_{g,S_{t,k-1},k}\right|\right)\varepsilon+\Delta_{s_{g,S_{t,k-1},k},S_{t,k'-1},k'}$$
$$\geq r(S_{t,k'-1}\cup\left\{s_{g,S_{t,k-1},k}\right\},\theta')+\Delta_{s_{g,S_{t,k-1},k},S_{t,k'-1},k'}-2B\left|\cup S_g'\right|\varepsilon$$
$$> r(S_{t,k'-1}\cup\left\{s_{g,S_{t,k-1},k}\right\},\theta')\,.$$

where the last inequality holds due to the requirement of $\varepsilon$ in Theorem 4. Above all, we conclude $\forall k'<k,s\notin S_{t,k'},r(S_{t,k'},\theta')>r(S_{t,k'-1}\cup\left\{s\right\},\theta')$, thus $S_{t,k-1}$ is still the set of units selected by $\texttt{Greedy}$ in the first $k-1$ steps under $\theta'$.

Next we prove that for any unit $s'\notin S_{t,k-1}\cup\left\{s_{g,S_{t,k-1}}\right\}$, $r(S_{t,k-1}\cup\left\{s'\right\},\theta')<r(S_{t,k-1}\cup\left\{s_{g,S_{t,k-1},k}\right\},\theta')$ holds.

$$r(S_{t,k-1}\cup\left\{s'\right\},\theta')=r(S_{t,k-1}\cup\left\{s'\right\},\theta_t)$$

$$\leq r(S_{t,k}, \theta_t) \quad \text{(Greedy's property)}$$

$$\leq r(S_{t,k}, \mu) + B \sum_{i \in \cup S_{t,k-1}} |\theta_{t,i} - \mu_i| + B \sum_{i \in s_{t,k}} |\theta_{t,i} - \mu_i| \quad \text{(Lipschitz continuity)}$$

$$\leq r(S_{t,k}, \mu) + B |\cup S_{t,k-1}| \varepsilon + B \sum_{i \in s_{t,k}} |\theta_{t,i} - \mu_i|$$

$$\leq r(S_{t,k}, \mu) + B |\cup S_{t,k-1}| \varepsilon + \Delta_{s_{t,k}, S_{t,k-1}, k} - B \left(2 |\cup S_{t,k-1}| + |s_{g,S_{t,k-1},k}| + 1\right) \varepsilon \tag{23}$$

$$\leq r(S_{t,k-1} \cup \{s_{g,S_{t,k-1},k}\}, \mu) + B |\cup S_{t,k-1}| \varepsilon - B \left(2 |\cup S_{t,k-1}| + |s_{g,S_{t,k-1},k}| + 1\right) \varepsilon \tag{24}$$

$$= r(S_{t,k-1} \cup \{s_{g,S_{t,k-1},k}\}, \mu) - B \left(|\cup S_{t,k-1}| + |s_{g,S_{t,k-1},k}| + 1\right) \varepsilon$$

$$\leq r(S_{t,k-1} \cup \{s_{g,S_{t,k-1},k}\}, \theta') - B\varepsilon \quad \text{(Lipschitz continuity)}$$

$$< r(S_{t,k-1} \cup \{s_{g,S_{t,k-1},k}\}, \theta'),$$

where the first equality is because $\theta'_i = \theta_{t,i}$ for any $i \in s'$ and $i \in \cup S_{t,k-1}$, the third inequality is because $\left\|\theta_{t,S_{t,k-1}} - \mu_{S_{t,k-1}}\right\|_\infty \leq \varepsilon$. (23) comes from the definition of $\neg D_k(t)$ and (24) is due to the definition of $\Delta_{s_{t,k}, S_{t,k-1}, k}$.

Above all, we have proved if event $\left\{S_{t,k-1} \in \sigma_{k-1}, \left\|\theta_{t,S_{t,k-1}} - \mu_{S_{t,k-1}}\right\|_\infty \leq \varepsilon, s_{t,k} \notin \sigma_k(S_{t,k-1}), \neg D_k(t)\right\}$ happens, then $\mathcal{E}_{S_{t,k-1},k,1}(t)$ holds.

Next we consider $\mathcal{E}_{S_{t,k-1},k,2}(t)$. By contradiction, when the event $\left\{S_{t,k-1} \in \sigma_{k-1}, \left\|\theta_{t,S_{t,k-1}} - \mu_{S_{t,k-1}}\right\|_\infty \leq \varepsilon, s_{t,k} \notin \sigma_k(S_{t,k-1}), \neg D_k(t)\right\}$ happens, if $\neg \mathcal{E}_{S_{t,k-1},k,2}(t) = \left\{\left\|\theta_{t,s_{g,S_{t,k-1},k}} - \mu_{s_{g,S_{t,k-1},k}}\right\|_\infty \leq \varepsilon\right\}$ holds, then $\theta_t$ satisfies the property of $\theta'$ defined in $\mathcal{E}_{S_{t,k-1},k,1}(t)$. Thus according to $\mathcal{E}_{S_{t,k-1},k,1}(t)$, $s_{g,S_{t,k-1},k}$ would be the $k$-th selected unit by Greedy when the input is $\theta_t$, or in other words $s_{t,k} = s_{g,S_{t,k-1},k} \in \sigma_k(S_{t,k-1})$. This contradicts $\left\{S_{t,k-1} \in \sigma_{k-1}, \left\|\theta_{t,S_{t,k-1}} - \mu_{S_{t,k-1}}\right\|_\infty \leq \varepsilon, s_{t,k} \notin \sigma_k(S_{t,k-1}), \neg D_k(t)\right\}$. Thus we conclude $\mathcal{E}_{S_{t,k-1},k,2}(t)$ also holds.

Above all, for term (22) we have

$$(22) \leq \mathbb{E}\left[\sum_{t=1}^T \mathbb{1}\left\{S_{t,k-1} \in \sigma_{k-1}, \mathcal{E}_{S_{t,k-1},k,1}(t), \mathcal{E}_{S_{t,k-1},k,2}(t)\right\}\right]$$

$$\leq \sum_{S \in \sigma_{k-1}} \mathbb{E}\left[\sum_{t=1}^T \mathbb{1}\{\mathcal{E}_{S,k,1}(t), \mathcal{E}_{S,k,2}(t)\}\right]$$

$$\leq \sum_{S \in \sigma_{k-1}} \sum_{q \geq 0} \mathbb{E}\left[\sum_{t=\tau_{S,k,q}+1}^{\tau_{S,k,q+1}} \mathbb{1}\{\mathcal{E}_{S,k,1}(t), \mathcal{E}_{S,k,2}(t)\}\right]$$

$$\leq \sum_{S \in \sigma_{k-1}} \sum_{q \geq 0} \mathbb{E}\left[\sup_{t \geq \tau_{S,k,q}+1} \prod_{i \in s_{g,S,k}} \frac{1}{\mathbb{P}\left(|\theta_{t,i} - \mu_i| \leq \varepsilon \mid \mathcal{H}_t\right)}\right] - 1$$

$$\leq \sum_{S \in \sigma_{k-1}} \left(\sum_{q=0}^{\lceil 8/\varepsilon^2 \rceil - 1} \left(c\varepsilon^{-4}\right)^{|s_{g,S,k}|} + \sum_{q \geq \lceil 8/\varepsilon^2 \rceil} e^{-\varepsilon^2 q/8} \left(c'\varepsilon^{-4}\right)^{|s_{g,S,k}|}\right)$$

$$\leq \sum_{S \in \sigma_{k-1}} \frac{C}{\varepsilon^2} \left(\frac{C'}{\varepsilon^4}\right)^{|s_{g,S,k}|},$$

where $\tau_{S,k,q}$ is the round at which $\mathcal{E}_{S,k,1}(t) \wedge \neg\mathcal{E}_{S,k,2}(t)$ occurs for the $q$-th time, note $\tau_{S,k,0} = 0$ for any $k \in [K], S \in \sigma_{k-1}$. The fourth inequality comes from the result in Lemma 2 and the fifth comes from the Lemma 8. Here $C, C'$ are two universal constants. $\qquad \square$

# F  Technical Lemmas

**Lemma 5.** *(Chernorff-Hoeffding bound) Let $X_1, X_2, \ldots, X_n$ be identical independent random variables such that $X_i \in [0, 1]$ and $\mathbb{E}[X_i] = \mu$ for any $i \in [n]$. Then for any $\epsilon \geq 0$, we have*

$$\mathbb{P}\left(\left|\frac{1}{n}\sum_{i=1}^{n}X_i - \mu\right| \geq \epsilon\right) \leq 2\exp\left(-2n\epsilon^2\right)$$

**Lemma 6.** *(Concentration and anti-concentration inequalities for Gaussian distributed random variables [1].) For a Gaussian distributed random variable $Z$ with mean $m$ and variance $\sigma^2$, for any $z$,*

$$\frac{1}{4\sqrt{\pi}}\exp\left(-\frac{7z^2}{2}\right) < \mathbb{P}\left(|Z - m| > z\sigma\right) \leq \frac{1}{2}\exp\left(-\frac{z^2}{2}\right).$$

**Lemma 7.** *(Lemma 3 in [30]) In Algorithm 2, for any base arm $i \in [m]$ and round $t$, we have*

$$\mathbb{P}\left(|\theta_{t,i} - \hat{\mu}_{t,i}| > \epsilon \mid a_{t,i}, b_{t,i}\right) \leq 2\exp\left(-2N_{t,i}\epsilon^2\right),$$

*where $a_{t,i}, b_{t,i}$ are the value of $a_i$ and $b_i$ before the start of round $t$.*

**Lemma 8.** *(Lemma 5 in [24]) In Algorithm 2, for any unit $s$, let $\tau'_q = \min\{t : N_{t,s} \geq q\}$, we have*

$$\mathbb{E}\left[\sup_{t \geq \tau'_q}\frac{1}{\prod_{i \in s}\mathbb{P}\left(|\theta_{t,i} - \mu_i| \leq \varepsilon \mid \mathcal{H}_t\right)}\right] - 1 \leq \begin{cases} \left(c\varepsilon^{-4}\right)^{|s|} & \text{for every } q \geq 0 \\ e^{-\varepsilon^2 q/8}\left(c'\varepsilon^{-4}\right)^{|s|} & \text{if } q > 8/\varepsilon^2, \end{cases}$$

*where $c$ and $c'$ are two universal constants.*