# OpenReview forum: "The Hardness Analysis of Thompson Sampling for Combinatorial Semi-bandits with Greedy Oracle"
_NeurIPS.cc/2021/Conference — NeurIPS 2021 Poster_

### Official Review · Reviewer_x5US · 2021-07-09

**Rating:** 7
**Confidence:** 3

**Summary:**

The paper presents a TS algorithm that uses Greedy oracle to minimise the cumulative regret on combinatorial multi-armed bandit (CMAB) setting. beside presenting an upper bound, they have also established a lower bound (on the cumulative regret) for their approach. The paper discuss the intuition of behind the proof of their proposed analysis for upper bound assuming the "Greedy" step (Line 4 in  Algorithm 1) returns a unique solution, and then in section 6.2 it relaxes that assumption.

CMAB is a well-known framework in the MAB literature, that generalises probabilistic maximum coverage (PMC), online influence
maximization (OIM), multiple-play MAB (MP-MAB), minimum spanning tree (MST) and many others. TS based regret-minimisation algorithms are very efficient for the family of single parameter reward distribution; there are recent studies that apply TS under the CMAB setup. Again, finding the optimal solution (in offline learning) for many problems (like PMC, OIM etc.) is NP-hard. On the other hand, Greedy oracle is a popular choice in various optimisation methods (including submodular maximisation).
Hence, the idea to attack the CMAB setup with TS using a Greedy oracle is very natural.

**Limitations And Societal Impact:**

Yes, the authors have discussed the limitation with clarity.

**Main Review:**

The paper is very well-written. The assumptions, problem statement, related work, and contributions are presented with great clarity. The upper and the lower bound (on the cumulative regret) presented in the paper are useful, and the authors have put a fair amount of discussion on the intuitions behind their approach.

The authors are being strongly recommended for presenting experimental results to compare with the existing approaches and validate the efficiency of their algorithm.


**Time Spent Reviewing:**

10

---

> ### Author Response · Authors · 2021-08-10
> **Reply to reviewer x5US**
>
> We thank the reviewer for the valuable recognition. Please find our following discussions.
>
> Since we mainly aim to tackle the theoretical convergence result for CTS with approximate oracles and there are many previous works [15,20,31,24,26] showing the efficiency advantages of TS-based algorithms, we do not include experiments. We will consider incorporating some experimental comparisons. Thanks for the suggestion.

---

### Official Review · Reviewer_Lbtx · 2021-07-09

**Rating:** 4
**Confidence:** 5

**Summary:**

The paper studies Thompson Sampling for combinatorial multi-armed bandit problems for based on a greedy oracle. An upper and lower bound on regret is provided.



**Ethical Concerns:**

No direct ethical concerns.

**Limitations And Societal Impact:**

Theoretical paper. No direct societal impacts.

**Main Review:**

Critique of the setting:

The paper only considers problems where the natural greedy algorithm (based on r(S,\mu)) is a good optimization algorithm. For some problems like the MNL-bandit (assortment optimization under the MNL) choice model [Agrawal et al 2018], the optimization algorithm is quite simple but not the natural greedy.

It's a bit odd that we are able to observe X_t,i for every base arm i. For example, in the lower bound instance, it is natural to be able to observe which nodes get influenced, as a result of the action set S, but unnatural to be able to observe which edge they got influenced through.

Regarding regret bound results:

The lower bound is only for the CTS algorithm and not for any bandit algorithm based on greedy algorithms. The intuition provided for the difficulty seems to hold for more general algorithm. Could the authors foresee a definition of more general algorithms that take incremental decisions like greedy and still face this lower bound?

Theorem 1 needs to be stated in terms of the general problem - not in terms of the instance parameters. How does the \Delta defined in this problem relate to Delta_min and Delta_max quantities? Is Delta = \Omega(Delta_min)? Is there a family of instances that can be defined to show that the regret lower bound changes with Delta and T in this way?

The lower bound vs upper bound comparison is quite unsatisfactory - it is a very specific case in which the two bounds are matching - it says nothing about the the dependence on K, unit sizes, B etc.  I think the authors should present the Theorem 1 instance as an example, rather than a lower bound. In any case I would suggest removing the claim of “almost matching upper and lower bound”.

How does the upper bound result translate to specific CMAB problems, and how does it compare to existing regret bounds for the problems studied previously in the literature?

Regarding proof techniques:

The proof is based on repeatedly applying the bound on number of exploration steps needed to select the right action at every step of the greedy algorithm. Since the upper bound the authors are aiming to prove is crudely log T/Delta^2 - with max over k,s a high probability bound on exploration steps suffices and detailed understanding of how much regret is lost is not required.

Further comments:

In the discussion below assumption 2, it will be useful to discuss what the value of B is for the different combinatorial bandit problems mentioned there.

Line 256: not sure what this means: “exploration on the denominator of..”. do you mean “on the order of”?

Summary:

Overall, the paper studies an interesting problem, but the upper bound seems to stop short of providing an interesting insight in the problem - both in terms of the actual form of the bound and proof techniques. The lower bound is not very detailed. It only shows a weak dependence on problem parameters for that specific instance.



**Time Spent Reviewing:**

3

---

> ### Author Response · Authors · 2021-08-10
> **Reply to reviewer Lbtx - Part 1**
>
> We thank  the reviewer for the valuable comments. Please find our clarifications below.
>
> -- "Only consider natural greedy algorithm": the significance of our result
>
> We agree that there are other oracles and greedy may not provide solutions for all combinatorial optimization problems. But in general, greedy is common to provide solutions for many problems including PMC, OIM, MP-MAB, and MST. These problems are widely studied as motivating examples in the CMAB literature [6,7,11,14,17,21,24,29,30,31,32].
>
> We would like to emphasize our contribution while explaining why we focus on the greedy oracle. The existing results for the analysis of the CTS algorithm need to assume exact oracles. And the work [30] shows a linear regret bound for an approximation oracle. Then there is a fundamental question of whether CTS cannot be used with approximation oracles. Since greedy oracle is one of the most common approximate oracles, we focus on it to give the first theoretical guarantee for CTS with approximate oracles. Our result shows that the linear regret example in [30] does not hold for any approximation oracle and breaks the misconception that CTS cannot be used with approximation oracles.
>
> -- "Observe $X_{t,i}$ for every base arm $i$": semi-bandit/edge-level feedback
>
> The feedback of observing each base arm in the selected action is semi-bandit feedback, which is well-studied in previous CMAB works [11,17,24,30,31]. For the PMC and OIM problems, such semi-bandit feedback is known as edge-level feedback, which is also well-studied in previous works [6,7,14,29,32]. The feedback "to observe which nodes get influenced" mentioned by the reviewer is known as node-level feedback and is also studied in some previous works [21; Vaswani et al, 2015]. Some work also discusses the differences between these two [Vaswani et al, 2015].
>
> Sharan Vaswani, Laks V.S. Lakshmanan, Mark Schmidt. Influence Maximization with Bandits. arXiv preprint arXiv:1503.00024, 2015.
>
> Taking PMC as an example. The input is a weighted bipartite graph $G=(L,R,E)$, where each edge $(u,v)$ is associated with a probability $p(u,v)$. The goal is to find a node set $S \subseteq L$ with $|S|=K$ to maximize the number of influenced nodes in $R$, where a node $v \in R$ can be influenced by node $u \in S$ with independent probability $p(u,v)$. This model can well characterize the scenario of advertisement placement: $L$ is the set of web pages, $R$ is the set of users, the probability $p(u,v)$ represents the probability that user $v$ clicks the advertisement on page $u$. The advertiser wants to repeatedly select a set of $K$ web pages to place an advertisement and maximize the number of users that have clicked on this advertisement displayed on any web page. Note in this case, if a user clicks on this advertisement, we can also observe on which web page he/she clicks, which corresponds to the edge-level/semi-bandit feedback.
>
> -- "The lower bound is only for the CTS algorithm and not for any bandit algorithm based on greedy algorithms. The intuition provided for the difficulty seems to hold for more general algorithm"
>
> We first would like to emphasize that our main contribution is the first to give the theoretical guarantee for CTS with a certain approximate oracle. We show a greedy regret bound of order $O(\log T/\Delta^2)$. Though the greedy regret is not $\alpha$-approximation regret, an upper bound for the former implies an upper bound for the latter. Also previous works on CUCB with exact/approximate oracles [29] and CTS with exact oracles [24,30] all have (approximate)-regret bound of order $O(\log T/\Delta)$.
>
> We agree that it is an open and interesting question whether $O(\log T/\Delta^2)$ is the best greedy regret achievable by any algorithm.
>
> A main reason to derive only greedy regret for CTS with a greedy oracle is that we find TS is not well-compatible with $\alpha$-approximation regret.
>
> Under UCB, since the monotonicity exists between the true parameter and the UCB parameter, the $\alpha$-approximate regret can be deducted as
> $$\alpha \cdot r(S*,\mu) - r(S_t,\mu) \le \alpha \cdot  r(S*,U) - r(S_t,\mu) \le r(S_t,U) - r(S_t,\mu) \le \sum_{i \in S_t} |U_i - \mu_i| \,.$$
> Thus it only needs to bound the number of selections of bad action $S_t$. UCB works nicely with the approximate regret. However under TS, since there is no monotonicity between the true parameter and the surrogate parameter, the approximate regret could only be deducted as
> $$\alpha\cdot r(S*,\mu) - r(S_t,\mu) \le \alpha\cdot r(S*,\mu) - \alpha \cdot r(S*,\theta) +\alpha \cdot r(S*,\theta) -  r(S_t,\mu) \le  \alpha\cdot r(S*,\mu) - \alpha\cdot r(S*,\theta) + r(St,\theta) - r(S_t,\mu)\le \alpha \sum_{i \in S*} |\theta_i - \mu_i| + \sum_{i \in S_t} |\theta_i - \mu_i|   \,.$$
> However, to bound this RHS, it requires a sufficient number of selections of the optimal action $S*$, which may not be the case with approximate oracles like the example in our Theorem 1.
>
> The $\alpha$-approximation regret is first brought up in analyzing UCB-based algorithms and may not well fit TS-based algorithms. We agree that after a different greedy regret is adopted, a further question is whether it makes any algorithm harder. We would leave this interesting question as future work.
>
> We would also leave the interesting question that "foresee a definition of more general algorithms that take incremental decisions like greedy and still face this lower bound" as future work.
>
> -- The relationship of $\Delta$ with $\Delta_{\min}$ and $\Delta_{\max}$
>
> There are no clear relationships between these terms. $\Delta_{\min}$ and $\Delta_{\max}$ are the minimum and the maximum reward gap from $S_g$ over all bad actions. However, $\Delta_{u,k}$ in Definition 1 is defined as the gap between $S_{g,k}$ and $S_{g,k-1}\cup$ { $u$ }. For $k\neq K$, our gap $\Delta_{u,k}$ is incomparable with the global defined gap $\Delta_{\min}$ and $\Delta_{\max}$, as the "optimal action" used to define $\Delta_{u,k}$ is $S_{g,k}$ while for the $\Delta_{\min}$ and $\Delta_{\max}$ is $S_g=S_{g,K}$. For $k=K$, we have $\Delta_{\min} \le \Delta_{u,K} \le \Delta_{\max}$ since the "optimal action" used to define these gaps is the same $S_g = S_{g,K}$.
>
> -- More general lower bound, dependence on $K$, unit sizes, and $B$
>
> We agree that it would be better to derive a general instance revealing dependence on other problem constants. However, the constructed instance has to satisfy some hard constraints.
> 1. The greedy solution $S_g$ has theoretical approximation guarantees but is NOT the optimal solution.
> 2. Some regret difference needs to be a constant, thus the numerator can not use a $\Delta$ to cancel $\Delta^2$ in the denominator.
>
> To make the solution a good combinatorial instance, the minimal value for $K$ is $2$. Even for $K=2$, we need to construct an instance and check the reward of every possible action to make sure the greedy output is not optimal. For the constant regret difference, we need to construct a unit $u$ such that the minimal regret difference of any action containing it is a constant. Though this way of construction may not be perfect, Theorem 1 gives a good instance to illustrate the hard situation $\Omega(\log T / \Delta^2)$ can happen. For general instances, it is hard to check greedy oracle is only approximate (not exact) and the existence of such a unit $u$. We would leave this an interesting future direction.
>
> We also would like to emphasize that even though this lower bound instance does not depend on some problem parameters, it does show the dependence of $\log T/\Delta^2$ is unavoidable. Note that all previous works on CUCB with exact/approximate oracles and CTS with exact oracles all have regret bound of order $O(\log T/\Delta)$. We give the first instance with lower bound on $\Omega(\log T/\Delta^2)$.

---

> ### Author Response · Authors · 2021-08-10
> **Reply to reviewer Lbtx - Part 2**
>
> -- How does the upper bound result translate to specific CMAB problems and how does it compare to existing regret bounds for the problems studied previously in the literature?
>
> To translate the upper bound result to specific CMAB problems, it only needs to replace $\Delta_{\max}$, $B$, $|s|$ in Eq(6) with their corresponding values of specific problems.
>
> Taking the PMC problem on bar graphs as an example. Here the bar graph is a special bipartite graph where each left node's outdegree is 1 (indegree is 0) and each right node's indegree is 1 (outdegree is 0). In this case, $B=1$, $|s|=1$ for any unit $s$, and $\Delta_{\max}=O(n)$, where $n$ is the number of nodes in each part and also the number of edges. So our greedy regret upper bound for this specific problem is $O(n^2 \log T/\Delta^2)$.
>
> This problem is also covered in the work [22]. Their greedy regret upper bound is $O(n^2 K \log T/\Delta^2)$, $O(K)$ worse than ours. Note even in this case, their algorithm needs to estimate $O(n\cdot 2^n)$ parameters, while we only need to estimate $O(n)$ parameters.
>
> This problem, if run using the CUCB algorithm, the approximate regret bound is $O(nK \log T/\Delta)$ [29], which is smaller than our CTS algorithm and the OG-UCB algorithm in [22].
>
> In general, the setting studied in [22] is not a CMAB setting, so we two are not directly comparable in general. Also, the work [24] is the most related work and is the state-of-the-art that studies CTS with exact oracles. Their regret upper bound is of order $O(\log T/\Delta)$. We are the first to analyze CTS with the approximate greedy oracles and the regret bound is of order $O(\log T/\Delta^2)$. Previous work [29] studying CUCB has proved the $\alpha$-approximate regret bound of order $O(\log T/\Delta)$.
>
> -- Proof techniques
>
> The proof technique is indeed not very difficult. However, we need to carefully design the event decomposition and adapt combinatorial constraints here. We would like to emphasize that we are the first to give a positive result on the convergence of CTS with a certain approximation oracle. All previous regret bounds for CTS need to work with exact oracles. We break the misconception that CTS cannot be used with approximation oracles.
>
> -- Value of $B$
>
> The value of $B$ has been discussed and computed in many previous works [6,7,29,32,21]. For the PMC, MP-MAB, and the OIM problem under the independent cascade model on bipartite graphs, the Lipschitz coefficient is $B = 1$. We will give some discussions about this in the revision.
>
> -- Line 256
>
> We mean $u_3$ needs to be explored/selected at least $\Omega(\log T/\Delta^2)$ times. However, for each selection/exploration, the algorithm needs to pay at least $(0.52\Delta+0.008)$ regret. Thus the gap $\Delta$ appearing in the denominator of $\Omega(\log T/\Delta^2)$ can not be canceled by this paid regret $(0.52\Delta+0.008)$ in the numerator. And thus the greedy regret is at least of order $\Omega(\log T/\Delta^2)$. We will revise this sentence to make it clearer.

---

### Official Review · Reviewer_kJUS · 2021-07-16

**Rating:** 5
**Confidence:** 3

**Summary:**

The paper studies the combinatorial semi-bandit problem. It provides the first analysis of Thompson sampling (TS) with the greedy oracle. It shows instances that the TS with Gaussian priors suffers $\Omega(\log T / \Delta^2)$ regret from. It proposes a modified TS with Beta priors and shows that the proposed algorithm achieves $O(\log T / \Delta^2)$ regret for the instances. The proposed algorithm enjoys near-optimal regret for the multi-armed bandit problem.

**Limitations And Societal Impact:**

The paper clarifies the assumptions of the theoretical analysis.

**Main Review:**

### Originality: Are the tasks or methods new? Is the work a novel combination of well-known techniques? (This can be valuable!) Is it clear how this work differs from previous contributions? Is related work adequately cited?

The paper provides the first analysis of TS with the greedy oracle. It compares the problem with a wide range of existing studies and confirms the novelty of the problem. The proposed algorithm is a simple and novel modification of TS.

### Quality: Is the submission technically sound? Are claims well supported (e.g., by theoretical analysis or experimental results)? Are the methods used appropriate? Is this a complete piece of work or work in progress? Are the authors careful and honest about evaluating both the strengths and weaknesses of their work?

The theoretical analysis looks correct. The paper shows regret lower bound of TS with Gaussian priors, but it shows upper bound of TS with Beta priors. It is open that the upper and lower bounds of TS with Gaussian and Beta priors, respectively.

### Clarity: Is the submission clearly written? Is it well organized? (If not, please make constructive suggestions for improving its clarity.) Does it adequately inform the reader? (Note that a superbly written paper provides enough information for an expert reader to reproduce its results.)

The paper is well written and easy to follow.

### Significance: Are the results important? Are others (researchers or practitioners) likely to use the ideas or build on them? Does the submission address a difficult task in a better way than previous work? Does it advance the state of the art in a demonstrable way? Does it provide unique data, unique conclusions about existing data, or a unique theoretical or experimental approach?

Analyzing algorithms with the approximation oracle is very important in practice. The paper motivates researchers to progress the theoretical analysis of TS with other approximation oracles.

================

After the discussion with other reviewers, I changed my score from 6 to 5.
I agree with the opinion that the regret lower bound is limited to a special instance and does not capture the essential difficulty of the problem.

**Time Spent Reviewing:**

3 hours

---

> ### Author Response · Authors · 2021-08-10
> **Reply to reviewer kJUS**
>
> We thank the reviewer for the valuable recognition. Please find our following discussions.
>
> -- Different priors
>
> The analysis for the upper bound with Gaussian prior is very similar to that with Beta prior, with only minor differences in the constant coefficients. We would also add this result in the revised version. The reason we use Beta prior in the upper bound is mainly that many combinatorial optimization problems have 0/1 binary feedback where Beta prior is more suitable. This usage is also adopted in many prior works of CMAB [30,17,14,24].
>
> To derive a lower bound for TS, a key property of anti-concentration is needed, where Gaussian distribution has a nice anti-concentration result. However, we are not aware of such a result for Beta distribution, even though we believe such a phenomenon of anti-concentration also exists for Beta distribution. Once a similar anti-concentration is proved for Beta distribution, the lower bound with Beta prior could be derived. We would leave this as an interesting future work.

---

### Official Review · Reviewer_1vtP · 2021-07-19

**Rating:** 4
**Confidence:** 3

**Summary:**

This paper studies TS for a certain set of Combinatorial semi-bandit problems with a greedy approximation algorithm.
It derives a logarithmic upper bound for this algorithm and also provides a single problem instance with $1/\Delta^2$ regret, where the upper bound is provably tight.

**Limitations And Societal Impact:**

Yes.

**Main Review:**

The problem setting seems relevant, though the authors do not do a very good job in describing the actual motivating problems. It would be good to formally define in a concise manner the OIM, PMC and MP-MAB problems at the beginning of the setting section.
It seems reasonable to define the regret against a baseline obtained by the offline oracle when finding the optimal solution is NP-hard.
However I see issues with the regret definition. To cite the authors: "The objective of the learning agent is to maximize the cumulative expected reward over $T$ rounds", which is not what the regret is actually capturing.
By defining the clipped per-step regret, we do not capture the cumulative expected reward over $T$ rounds anymore.

I did not check the details of the upper bound proof in the appendix, but the result sounds reasonable. It is at first surprising that the regret scales with $1/\Delta^2$ instead of the typical $1/\Delta$. The authors show in the lower bound that this squared dependency is unavoidable for this algorithm and is due to the fine workings of the greedy oracle. The regret definition with the clipped per-step regret serves as a short-cut to the lower bound proof. The proof should also go through with an unclipped regret but requires more technical work.

Overall I am not convinced how meaningful regret bounds in this setting are. The actual performance is dominated by the effect of the greedy oracle. Swapping the means of $u_1$ and $u_4$ in the example changes the regret of the algorithm from $1/\Delta^2$ to a finite value, even though the actual performance compared to the optimal action is almost identical.
The presented bounds are apparently worst-case bounds and the gap definitions do not capture the inherent hardness of the problem very well.

Minor: Removing the redundant use of the word "could" would improve the reading flow.



**Time Spent Reviewing:**

2

---

> ### Author Response · Authors · 2021-08-10
> **Reply to reviewer 1vtP**
>
> We thank the reviewer for the valuable comments. Please find our clarifications below.
>
> -- Motivating example
>
> Since OIM, PMC and MP-MAB are common examples in CMAB [6,7,11,14,17,21,24,29,30,31,32] and the space is limited, we do not include a too detailed introduction about these models. To make our work more self-contained, we will describe the fittest model PMC in more detail in the revision.
>
> The input for the PMC problem is a weighted bipartite graph $G=(L,R,E)$, where each edge $(u,v)$ is associated with a probability $p(u,v)$.  The goal is to find a node set $S\subseteq L$ with $|S|=K$ to maximize the number of influenced nodes in $R$, where a node $v \in R$ can be activated by node $u \in S$ with independent probability $p(u,v)$. We will add this description in the beginning of the setting section.
>
> Thanks the reviewer for the suggestion.
>
> -- Clipped per-step regret
>
> We claim the clipped per-step regret is a very natural definition in the setting of approximation oracles.
>
> Even though there is no such max(-,0) in the definition of [22] and other works with approximation regret [6,7,29], all these regret upper bound analyses do relax the per-round regret to max(-,0) since it is easier to mainly bound the number of selections on "bad actions".
>
> This clipped per-step regret definition mainly serves in the analysis of the lower bound. If we do not add such max(-,0), then one also needs to bound the number of selections on "good actions" and minus their reward contributions from the regret. We do not know how to bound this for the problem-dependent bound. In our knowledge, most regret analyses focus to bound the number of selections on "bad actions". Perhaps some adversarial algorithms can deal with this, but they only provide problem-independent (not problem-dependent) regret bound. Also, perhaps there is some way to work around this, but we are not sure how "the proof should also go through with an unclipped regret but requires more technical work".
>
> For the lower bound in [22], though they do not explicitly have such max(-,0), the greedy oracle in that class of instances is in fact exact and the per-round regret is always nonnegative.
>
> Our statement "The objective of the learning agent is to maximize the cumulative expected reward over T rounds, or equivalently to minimizing the cumulative expected regret..." is not very accurate. We will revise this sentence. Thanks for the suggestion.
>
> -- "The actual performance is dominated by the effect of the greedy oracle": Significance of the result
>
> The existing results for the analysis of the CTS algorithm need to assume exact oracles. And the work [30] shows a linear regret bound for an approximation oracle. Then there is a fundamental question of whether CTS cannot be used with approximation oracles. Since greedy oracle is one of the most common approximate oracles, we focus on it to give the first theoretical guarantee for CTS with approximate oracles. So yes, our work indeed focuses on the greedy oracle and we think it has valuable merits. Our result shows that the linear regret example in [30] does not hold for any approximation oracle and breaks the misconception that CTS cannot be used with approximation oracles.
>
> -- "even though the actual performance compared to the optimal action is almost identical"
>
> We are not sure what does this "the actual performance compared to the optimal action is almost identical" mean. Does the reviewer mean these two cases have similar performances (or upper bound) but drastically different lower bounds?
>
> Usually, two similar instances do not necessarily have similar regret results. Consider two MAB instances of two arms with mean $(1, 1-\Delta)$ and mean $(1,1)$. The problem-dependent regret bound of the former is $O(\log T / \Delta)$, which is larger if $\Delta$ is small. However, the performance of the former is more "identical" to that of $(1,1)$ if $\Delta$ is small.
>
> -- "The presented bounds are apparently worst-case bounds and the gap definitions do not capture the inherent hardness of the problem very well"
>
> Does the reviewer mean "worst-case bounds" are not very satisfactory? Usually, most lower bound instances are worst-case instances. Would the reviewer like to have the lower bound depending on the instance structure? We know some work on graph feedback (listed below) does have the kind of lower bound depending on graph properties, but their bound is problem-independent and free of the means.
>
> Noga Alon, Nicolò Cesa-Bianchi, Ofer Dekel, Tomer Koren. Online Learning with Feedback Graphs: Beyond Bandits. Proceedings of The 28th Conference on Learning Theory, PMLR 40:23-35, 2015.
>
> As pointed out by the reviewer, "swapping the means of $u_1$ and $u_4$ in the example changes the regret of the algorithm from $1/\Delta^2$ to a finite value", which states that our lower bound depends on the mean values. This is not special. For example, in graph bandits, their problem-dependent lower bounds (listed below) are solutions to optimization problems that depend on both the graph structure and the mean values.
>
> Yifan Wu, András György, Csaba Szepesvari. Online Learning with Gaussian Payoffs and Side Observations. Advances in Neural Information Processing Systems. 2015.
>
> Shuai Li, Wei Chen, Zheng Wen, Kwong-Sak Leung. Stochastic Online Learning with Probabilistic Graph Feedback. The 34th AAAI Conference on Artificial Intelligence (AAAI), 2020.
>
> The lower bound problem is even more difficult in our combinatorial setting since the constructed combinatorial instance has to satisfy some hard constraints.
> 1. The greedy solution $S_g$ has theoretical approximation guarantees but is NOT the optimal solution.
> 2. Some regret difference needs to be a constant, thus the numerator can not use a $\Delta$ to cancel $\Delta^2$ in the denominator.
>
> To make the solution a good combinatorial instance, the minimal value for $K$ is $2$. Even for $K=2$, we need to construct an instance and check the reward of every possible action to make sure the greedy output is not optimal. For the constant regret difference, we need to construct a unit $u$ such that the minimal regret difference of any action containing it is a constant. Though this way of construction may not be perfect, Theorem 1 gives a good instance to illustrate the hard situation $\Omega(\log T/\Delta^2)$ can happen.
>
> For general lower bound instances, it is hard to check greedy oracle is only approximate (not exact) and the existence of such a unit $u$. We would leave this as an interesting future direction.
>
> -- Minor comment
>
> Thanks for the suggestion. We will polish our writing presentations.

---

### Decision · Program_Chairs · 2021-09-28

**Decision:**

Accept (Poster)

**Comment:**

While the review team recognizes the importance of the problem and like the direction that the paper took, they also felt that the paper is not yet ready for publication. The main concerns are:

* model is not well motivated. In some problems (like OIM) the feedback assumed here is inconsistent with what is observed in practice (nodes vs edges).
* the regret bounds don't directly reflect the performance of the algorithm
* lower bound is too stylized and may not say much about the actual hardness of the problem

While the reviewers appreciate the author responses on those issues, they were not sufficient to significantly change the sentiment.

**Consistency Experiment:**

NeurIPS has a long history of experimentation. In 2014, NeurIPS ran an experiment in which 10% of submissions were reviewed by two independent committees to quantify the randomness in the review process. This year, we repeated a variant of this experiment to see how the quality of the review process has changed over time.  This paper was part of the experiment and was therefore assigned to two committees (consisting of reviewers, an Area Chair, and a Senior Area Chair) that reached independent decisions.  If both committees made the same recommendation, this recommendation was followed. If a single committee recommended acceptance, the paper was accepted (with the exception of a few cases in which the other committee identified what we considered a fatal flaw, e.g., an error in a key result).

This copy’s committee reached the following decision: **Reject**

The other committee assigned to the paper recommended **Accept (Poster)**.  You can find the other set of reviews, along with any follow up discussion with the authors here:
https://openreview.net/forum?id=ulqMdBThHsC